# Signal Dynamics in Diffusion Models: Enhancing Text-to-Image Alignment through Step Selection

## Abstract

Visual generative AI models often encounter challenges related to text-image alignment and reasoning limitations. This paper presents a novel method for selectively enhancing the signal at critical diffusion steps, optimizing image generation based on input semantics. Our approach addresses the shortcomings of early-stage signal modifications, demonstrating that adjustments made at later stages yield superior results. We conduct extensive experiments to validate the effectiveness of our method in producing semantically aligned images, achieving state-of-the-art performance. Our results highlight the importance of a judicious choice of sampling stage to improve diffusion performance and overall image alignment.[1]

## 1 Introduction

Visual Generative AI Models usually rely on diffusion models (Ho et al., 2020) that are conditioned by a textual prompt to guide the diffusion process during the inference, resulting in visually pleased images (Rombach et al., 2022; Podell et al., 2023; Ramesh et al., 2022; Saharia et al., 2022). Although these models show impressive semantic and compositional capacities, even the best models still suffer from text-image alignment and reasoning limitations (*e.g.* spatial, counting). Some works address these issues by improving the noisy captions in training datasets (Chen et al., 2023; 2024a; Segalis et al., 2023) or improving the architecture (Peebles & Xie, 2022), while others adopt an attention-based Generative Semantic Nursing (GSN) approach (Chefer et al., 2023; Rassin et al., 2023; Guo et al., 2024) that avoids retraining the whole model by correcting it at inference or adding conditioning to better guide the generation.

| Catastrophic Neglect | Subject Mixing | Attribute Binding and Leaking |
| --- | --- | --- |
| a photo of a giraffe and a banana | a photo of a giraffe and a bear | a photo of a car and a blue cat |

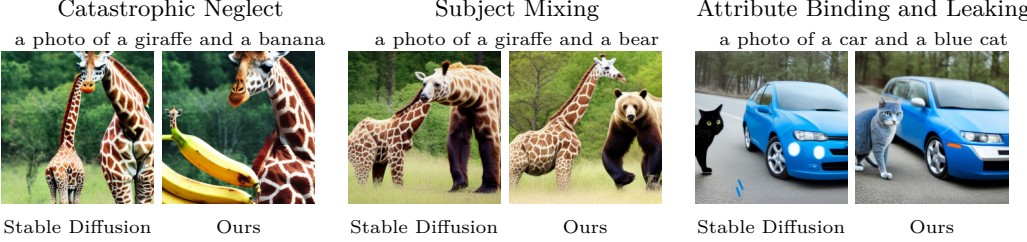

| Stable Diffusion | Ours | Stable Diffusion | Ours | Stable Diffusion | Ours |

Figure 1: Comparison of samples generated by Stable Diffusion and Ours.

Early research has identified several text-image alignment issues (Ramesh et al., 2022; Saharia et al., 2022; Chefer et al., 2023; Feng et al., 2023). These issues (Figure 1) include *catastrophic neglect*, where one or more elements of the prompt fail to be generated; *subject mixing*, where distinct elements are improperly combined; *attribute binding*, where attributes (*e.g.* color) are incorrectly assigned to the wrong entities while neglecting the correct ones; and *attribute leaking*, where attributes are correctly bound to the specified elements but are erroneously applied to additional, unintended elements in the scene.

---

[1]The code will be publicly released.

To improve generation, training-free methods (Chefer et al., 2023; Rassin et al., 2023; Li et al., 2023b; Guo et al., 2024; Agarwal et al., 2023) have emerged. These methods leverage the text-image relationship in the models diffusion features to optimize the latent image that the diffusion model is denoising to adjust it. However, these approaches require testing and carefully selecting multiple sensitive hyperparameters (*e.g.* choosing various diffusion steps to perform optimization or setting different loss thresholds to reach for each diffusion step), which can lead to potential failures during the optimization process. In addition, although multiple refinement steps are commonly employed along the diffusion path, the necessity for their repeated use and the reasoning behind their placement have been determined largely through experimental results, without clear explanation. We argue that a closer examination of the location of refinement steps would not only improve performance but also provide a better understanding of the optimal location of these steps. To mitigate the risk of under/over optimization, InitNO (Guo et al., 2024) optimizes multiple initial latent images solely at the first diffusion step, where latent images are pure Gaussian noise. However, the diffusion models reverse process reconstructs the signal gradually during image generation, making early-stage optimization less effective due to the weak signal at that point. As the signal becomes stronger in later diffusion steps, it provides more useful information for the refinement of the latent image. A deeper understanding of signal degradation dynamics can be used to improve generation capacity. In this work, we examine the impact of selecting the optimal diffusion steps to enhance the signal based on semantic content and demonstrate that carefully selecting these steps leads to substantial improvements in text-to-image alignment.

Our main contributions are the followings: 1) We propose a method for selectively enhancing the signal at a key diffusion step, optimizing image generation based on the input semantics. 2) We demonstrate that early-stage signal modification is less effective and show that later adjustments lead to improved results. 3) We validate our approach through extensive experiments, demonstrating its effectiveness in producing semantically aligned images and achieving state-of-the-art results while also studying the placement of the refinement steps.

## 2 Related work

**New controls and GSN** Li et al. (2023a) and Mou et al. (2023) introduce trainable modules to enable the addition of new conditions to the frozen models. Similarly, Zhang et al. (2023) incorporate a trainable copy of the model that can be conditioned on various control inputs, such as a drawing, a bounding box, or a depth map. Recent research focuses on conditioning models by working on the noisy latent image. SDEdit (Meng et al., 2022) adds varying levels of noise to an image, balancing between fidelity to the original image and creative variation. Sun et al. (2024) create pseudo-guide images by placing objects on a background, adding noise, and then denoising them to maintain object placement during generation. Choi et al. (2021) inject down-sampled guide images during diffusion to create variations of the guided image.

Generative Semantic Nursing (GSN) was introduced by Attend&Excite (Chefer et al., 2023), aiming to optimize the latent image during inference to better consider semantic information without having to retrain models. The latent image $x_t$ at step $t$ is modified by applying gradient descent step w.r.t a loss $\mathcal{L}$ on the extracted features produced by the model with the input $x_t : x_{t'} \leftarrow x_t - \alpha_t \cdot \nabla_{x_t}\mathcal{L}$ ($\alpha_t$ the learning rate). Hence, it shifts the latent image to achieve the objective conceptualized by the loss function. Attend&Excite considers the cross-attention features, which establish a link between image and text features, to ensure that the model adequately generates the subjects in the prompt. Building upon this approach, Syngen (Rassin et al., 2023), Divide and Bind(Li et al., 2023b), InitNO (Guo et al., 2024) and A-Star (Agarwal et al., 2023) design other loss functions to better enhance the alignment of the prompts while Chen et al. (2024b); Xie et al. (2023) combine layout information to textual information to force objects placement. The closest work to ours is InitNO, which performs a warm-up multi-round optimization on the initial latent image (initial noise). That is, they attempt to shift the initial latent image to reach a desired loss score, aiming to find an initial noise that will perform better during the generation process. The term "multi-round" applies because this process can take up to five rounds if the target loss score is not met, with a new initial latent image being resampled and optimized each

time. In contrast, we argue that the optimization of the latent image is more effective at a later step than at the initial step. As the partial information of the latent image becomes progressively more accurate, it is beneficial to refine the information at a distant step, where the latent image is easier to distinguish from the noise, where the diffusion has a more accurate understanding of the signal in the latent image. In addition, our method is more efficient without the use of multi-round optimization.

**Signal leak in diffusion models**  Lin et al. (2024) reveal that Stable Diffusion 1.4 and some other diffusion models exhibit signal leakage, meaning the signal does not completely vanish even in the final steps of the forward process. Everaert et al. (2024) exploit this signal leakage to gain control over the generated images, biasing the generation towards desired styles, enhancing image variety, and influencing colors and brightness. Grimal et al. (2024) demonstrate that certain noises during inference perform better for generating multiple objects. We hypothesize that this performance arises from a signal in the initial noise, which is more consistent to make multiple objects appear. Based on the signal construction during the denoising, we identify the diffusion step where we can improve the signal and align it with the text.

## 3 METHODOLOGY

### 3.1 PRELIMINARY: DIFFUSION MODELS

Diffusion models involve two processes: a forward process $q$ that progressively degrades images, and a reverse process $p$ that iteratively removes noise by retracing the forward steps. In this work, we adopt the variance-preserving approach from the Denoising Diffusion Probabilistic Model (DDPM) (Ho et al., 2020) in discrete time. The forward process is a Markov chain of length $T$ that adds small Gaussian noise to the data, described by $q(x_t|x_{t-1})$, and ultimately results in $x_T \sim \mathcal{N}(0, I)$. This process can be reparameterized as $q(x_t|x_0)$ to estimate any $x_t$ directly from $x_0$:

$$x_t = \sqrt{\bar{\alpha}_t}x_0 + \sqrt{1 - \bar{\alpha}_t}\epsilon, \quad \text{where } \epsilon \sim \mathcal{N}(0, I) \tag{1}$$

which can be interpreted as an interpolation between the signal $x_0$ and the noise $\epsilon$ . The noise scheduler defines the predetermined variance schedule, and consequently, the value of $\bar{\alpha}_t$, which determines how the signal $x_0$ will be degraded. As $\bar{\alpha}_t$ increases, the signal becomes harder to distinguish from the noise.

A neural network $p_\theta$ learns the reverse process. The model can be reparameterized in $\epsilon_\theta$ to predict directly the added noise with the corresponding objective:

$$L = \mathbb{E}_{x, \epsilon \sim \mathcal{N}(0, I), t}\Big[\|\epsilon - \epsilon_\theta(\mathbf{x}_t, t)\|^2\Big] \tag{2}$$

To condition the generation with text, the models of Rombach et al. (2022); Podell et al. (2023); Saharia et al. (2022); Chen et al. (2023); Balaji et al. (2023) adopt a cross-attention mechanism consisting of using the embedding of a prompt $p$ from a frozen textual encoder $\tau(\cdot)$ like T5 (Raffel et al., 2020) or CLIP (Radford et al., 2021). The textual encoder generates an embedding of $N$ tokens, which the model utilizes across different cross-attention layers. Within these layers, a linear projection is applied to the intermediate features $Q$ and the text embedding $K$. Attention maps are then computed as $A = \text{softmax}(QK^T/\sqrt{d})$. These attention maps can be reshaped into $\mathbb{R}^{h \times w \times N}$, where $h$ and $w$ represent the dimensions of the attention maps in the cross-attention layer, and $N$ denotes the sequence length of the prompt embedding. As demonstrated by (Hertz et al., 2022; Tang et al., 2023), cross-attention maps reveal meaningful semantic relationships between the spatial layout and corresponding words, which can be utilized for visualization and control. With the text-conditioning, the training objective becomes:

$$L = \mathbb{E}_{x, p, \epsilon \sim \mathcal{N}(0, I), t}\Big[\|\epsilon - \epsilon_\theta(\mathbf{x}_t, t, \tau(p))\|^2\Big] \tag{3}$$

To reduce the computational cost of diffusion, Rombach et al. (2022) developed a Latent Diffusion Model (LDM) that operates within a smaller perceptual latent space. This model

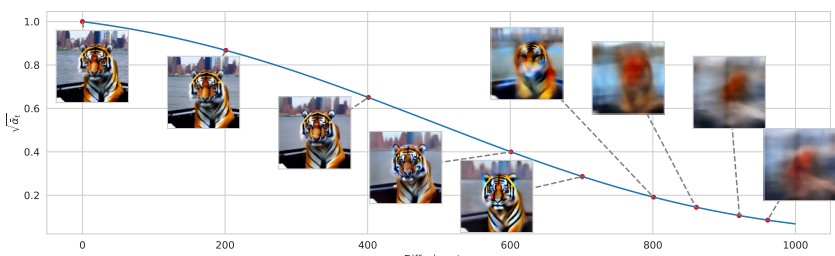

Figure 2: Value of $\bar{\alpha}_t$ as a function of the diffusion step $t$. The estimated $\hat{x}_0$ during the generation of "a photo of a tiger on a boat arriving in new york" at various steps is displayed. A coarse-to-fine generation is observed; as the denoising process progresses, the scene becomes increasingly distinguishable. Generate with Stable Diffusion 1.4.

generates an initial latent noise $z_T$, denoises it iteratively to obtain $z_0$, and then projects the latent image into pixel space to produce the final image $x_0$. Although our experiments use an LDM, our approach is equally applicable in pixel space. For clarity, we will describe the method using $x_t$, even though our experiments are conducted in the latent space.

During inference, we can generate an image without following the full training steps by using a sampling scheduler that discretizes the diffusion process into a reduced number of steps. For example, with the DDPM scheduler and 50 sampling steps, the first sampling step 0 corresponds to step 981 of the original diffusion process, significantly reducing the number of steps required while maintaining generation quality. To improve the process, recent approaches adopt two processes that can be combined but have different purposes. First, they adopt *GSN guidance* (GSNg) such that the latent image $x_t$ at step $t$ is shifted by applying a (unique) gradient descent step w.r.t a loss $\mathcal{L}$ that favors the alignment with the prompt, thus $x_t : x_{t'} \leftarrow x_t - \alpha_t \cdot \nabla_{x_t}\mathcal{L}$, with $\alpha_t$, the learning rate. Second, the process can be repeated at each of some predefined diffusion steps $t_1 \dots t_k$ until either $\mathcal{L}$ reach sufficient threshold or a specified maximum number of shifts has been made. This process is called *iterative refinement* (IterRef) step. We argue that choosing carefully the step at which IterRef is performed allows us to do it once only, without needing to compare to a threshold, thus reducing the number of hyperparameters to set while leading to better results.

### 3.2 Choose the right IterRef steps to enhance the content

Our method focuses on selecting the appropriate diffusion steps to enhance the signal within the noise, thereby generating more faithful final images. Previous studies have shown the coarse-to-fine behavior of diffusion models (Park et al., 2023). During the reverse process, the model reconstructs the low-frequency structure of the image first, before progressively refining the fine details towards the end. This behavior can be understood from the interpolation in Equation 1, where, as the forward process progresses, the signal $x_0$ diminishes while the noise $\epsilon$ increases. Importantly, at each diffusion step, we can estimate the final image and obtain an approximation of the underlying signal. Given any $x_t$ at a particular diffusion step, the final image $x_0$ can be estimated as:

$$\hat{x}_0 = (x_t - \sqrt{1 - \bar{\alpha}_t}\epsilon_\theta(x_t))/\sqrt{\bar{\alpha}_t} \tag{4}$$

In Figure 2, we visualize the estimated signal during the diffusion process, alongside the value of $\sqrt{\bar{\alpha}_t}$. As the process progresses, the signal becomes more defined, allowing the general structure of the final image to emerge even in the early stages. The degradation and reconstruction of the signal $x_0$ are controlled by the noise scheduler. Previous studies (Choi et al., 2022; Chen, 2023) have emphasized the importance of carefully selecting the noise schedule to allocate sufficient time for the model to construct the main content of the image. This ensures that the model has ample opportunity to build the scene accurately. In the context of semantic image generation, this explains why attention to the text prompt is stronger at earlier noise levels when the core elements of the image are still being

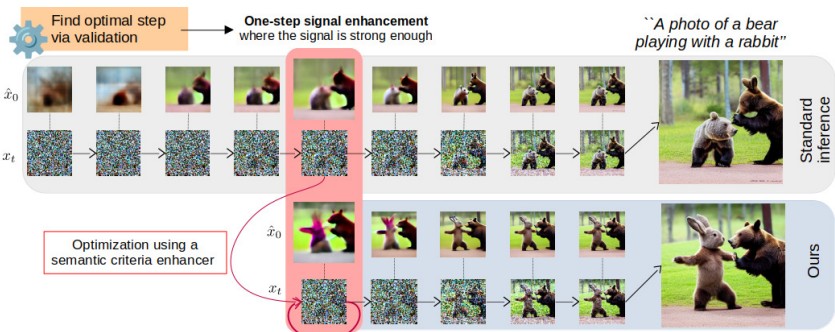

Figure 3: The diffusion process is paused at a key step (determined on a validation subset) to enhance the signal in the latent image. By amplifying the signal at this critical point, we ensure that the model can correctly construct the main components of the image, leading to a more accurate final result.

formed (Balaji et al., 2023; Park et al., 2023). At later stages, text input has less influence, the model focusing on refining details while the general spatial structure remains the same.

Our method leverages this understanding by enhancing the signal at the critical diffusion steps: neither too early, when the signal is weak, nor too late, when the scene is already defined. This ensures that the signal remains sufficiently strong throughout the reverse process, guiding the model to semantically construct the final image accurately. By carefully choosing the step, we can amplify the signal in the latent image, allowing for better semantic alignment with the text prompt. To select the best-performing steps automatically, we propose a validation method to test multiple steps on an evaluation metric (see 4.1). Our approach is summarized in Figure 3.

Since the latent space of diffusion models inherently lacks semantic meaning (Kwon et al., 2023; Park et al., 2023), making it unsuitable for direct manipulation to control the generated results, we rely on the model's ability to interpret the latent representation to assign semantic relevance and we use a single IterRef step to enhance the signal and ensure faithful alignment between text and image. In other words, we modify the signal interpreted by the model to enhance its quality, ensuring that the model receives an appropriate signal for a correct generation. Additionally, our only-one IterRef step approach is versatile and can be integrated with methods like GSNg for further improvement in image generation.

### 3.3 ENHANCE THE SIGNAL ACCORDING TO THE TEXT-TO-IMAGE ALIGNMENT TASK

Considering a prompt $p$ with a list of subject tokens $\mathcal{S} = \{s_1, \ldots, s_k\}$, we extract attention features for each subject. Following (Chefer et al., 2023), we use the cross-attention maps from the resolutions $16 \times 16$ pixels, averaging across heads and layers, followed by Gaussian smoothing. This results in a set of attention maps $A \in \mathcal{R}^{16 \times 16 \times n}$ with $n$ number of tokens (more details in Appendix). To encourage, we combine two losses. To ensure an attention for each subject token, we leverage the criterion from (Chefer et al., 2023):

$$\mathcal{L}_{CN} = \max_{s \in S}(1 - \max_{i,j}(A_{i,j}^s)) \tag{5}$$

where $A_{i,j}^s$ represents the cross-attention value at position $i, j$ for the subject token $s$. This loss encourages the token with minimal activation to be more excited. Additionally, we implement an Intersection Over Union (IoU) loss, already used in (Agarwal et al., 2023), to mitigate catastrophic mixing by fostering subject separation. For all combinations of subject token pairs $\mathcal{C}$, the loss is defined as:

$$\mathcal{L}_{\text{IoU}} = \frac{1}{|\mathcal{C}|} \sum_{\forall (m,n) \in \mathcal{C}} \left( \frac{\sum_{i,j} \min(A_{i,j}^m, A_{i,j}^n)}{\sum_{i,j}(A_{i,j}^m + A_{i,j}^n)} \right) \tag{6}$$

Table 1: Overview of methods. Steps are given in terms of sampling scheduler. *Max Shift* indicates the maximum predefined shifts applied if no threshold is met or if no threshold is used. *Max Gradient Updates* refers to the maximum number of times the latent image is updated during the generation.

| Methods | IterRef Which Step | IterRef Reach Threshold | IterRef Max Shift | GSNg | Max Gradient Updates of $x_t$ |
|---|---|---|---|---|---|
| Syngen | ø | ø | ø | 25 first steps | 25 |
| Attend&Excite | 0 10 20 | ✓ | 20 | 25 first steps | 85 |
| Divide&Bind | 0 10 20 | ✓ | 50 | 25 first steps | 175 |
| InitNO | 0 | ✓up to 4 restart if it fails | 50 | ø | 250 |
| InitNO+ | 0 / 10 20 | ✓up to 4 restart if it fails / ✓ | 50 / 20 | 25 first steps | 315 |
| Ours | 8 | ø | 50 | ø | 50 |
| Ours+ | 2 | ø | 50 | from 3 to 25 | 73 |

where $A_{i,j}^s$ denotes the cross-attention value at position $i, j$ for subject token $s$. In summary, our joint loss is defined as $\mathcal{L} = \mathcal{L}_{CN} + \mathcal{L}_{\text{IoU}}$, which we minimize using 50 shifting steps of the latent image $x_t$ with the Adam optimizer (Kingma & Ba, 2017) and a learning rate of $1 \times 10^{-2}$. These hyperparameters are fixed according to previous studies for a fair comparison.

## 4 Empirical Analysis and Results

### 4.1 Experimental settings

**Implementations** We mainly use Stable Diffusion 1.4 (SD 1.4) as all hyperparameters methods are based on this model. Images are generated using the DDPM Scheduler with 50 inference steps, on an Nvidia A100 80GB in Float 32 precision, with a Classifier-Free Guidance (Ho & Salimans, 2022) of 7.5. We compare our approach against the standard inference of Stable Diffusion, Attend&Excite, Divide&Bind (Li et al., 2023b), InitNO, and Syngen. We exclude A-Star due to a lack of an official implementation and because InitNO reports superior results. The authors of InitNO propose to couple their methods with GSNg and IterRef steps, which we refer to as InitNO+. We refer to our method as Ours and its variant incorporating the GSNg from Syngen as Ours+, where the GSNg is applied after the iterative refinement step. We also compare the results of Stable Diffusion 3 (SD 3) (Esser et al., 2024) with and without our approach. We summarize the differences between the methods in Table 1 and give more details on each method in the Appendix.

**Evaluation** To estimate prompt-image alignment, we report the TIAM score (Grimal et al., 2024), which assesses the model's ability to generate requested entities. The score reflects the proportion of correctly generated images. Following the recommended sampling method, we generated prompts for all possible combinations of two and three subject entities using 24 COCO labels (Lin et al., 2014). Each prompt generated multiple images, which were automatically evaluated to ensure the correct appearance of the requested entities and, where applicable, their attributes such as color. We created four datasets: two entities, two colored entities, three entities, three colored entities. For each dataset, 300 prompts were sampled, and 16 images per prompt were generated using the same 16 seeds to create the test set. In addition, we create four validation datasets by sampling 10 prompts, different from the 300 ones, that are used to determine the suitable IterRef step. We compute the CLIP Score (Radford et al., 2021) to measure the average alignment between text and image embeddings. Additionally, we employ the CLIP-based metrics proposed by Chefer et al. (2023), referred in this paper as the Similarity Score, which includes Full Prompt Similarity, Minimum Object Similarity, and Text-Text Similarity. However, caution is necessary when using CLIP-based metrics, as they often struggle with relational understanding, can misassociate objects with their attributes, and exhibit a significant lack of order sensitivity(Yuksekgonul et al., 2023). Finally, we use

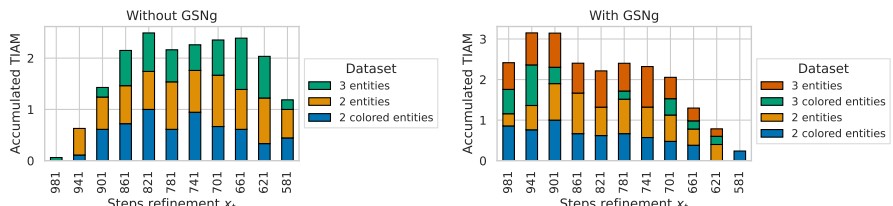

Figure 4: Accumulated TIAM scores without (left) and with (right) GSNg. The dataset with three colored entities is excluded on the left due to its low scores. Steps 821 and 941 are identified as optimal.

LAIONs aesthetic predictor[2] (Schuhmann et al., 2022) to estimate aesthetic quality on a scale from 1 to 10. Further details about the evaluation metrics in the Appendix.

**Optimal IterRef step selection** To select the optimal IterRef step, we evaluate 11 sampling steps, spaced every two steps (*i.e.* 0, 2, 4,..., 24) out of the 50 sampling steps for SD 1.4. We focus on the first 25 steps, as prior research shows limited benefit beyond this point (Chefer et al., 2023). For each validation dataset, we generate 16 images per prompt using the same 16 seeds and compute the TIAM score. We standardize the scores using a min-max scaler for each dataset and present the *accumulated standardized TIAM* across the IterRef step for Ours and Ours+ in Figure 4. Based on the scores, we find that step 821 (sampling step 8) is optimal without GSNg, while step 941 (sampling step 2) produces better results with GSNg. This difference can be explained by the need for changes to occur later in the process without GSNg, ensuring that the adjusted signal is strong enough to persist through random sampling. In contrast, GSNg enables continuous signal refinement, allowing for corrections even at later stages. Moreover, we calculate the aesthetic score and observe no degradation whatever the IterRef step chosen, confirming the choice (values available in the Appendix). We will use these selected steps in subsequent experiments. Our validation method is computationally efficient, requiring only 10 prompts with a limited number of samples to determine the optimal IterRef step. We applied the same approach to select the best step for SD 3 (details in Appendix).

### 4.2 RESULTS

#### 4.2.1 QUANTITATIVE RESULTS

**TIAM** We present in Table 2 the TIAM, CLIP and aesthetics scores of our method and other approaches. With SD 1.4, our method outperforms InitNO across all configurations in both TIAM and CLIP scores, using a single IterRef step without GSNg. This indicates that a single IterRef step is more effective when the signal is stronger than at the initial diffusion step, as expected by our approach. When combined with GSNg, we surpass all other methods in terms of TIAM scores, showing that GSNg leads to better results with our carefully chosen IterRef step. We achieve superior CLIP scores in all configurations, except for the three colored entities, where TIAM alignment scores are generally very low across all methods. For a fair comparison, we tried to add an IterRef step for the Syngen approach, referred as Syngen+, but obtained an even lower score. More details in the Appendix. With SD 3, our method mitigates catastrophic neglect, showing improved TIAM and CLIP scores for two and three entities. We note a slight decrease in performance for two colored entities but nearly identical TIAM scores with a better CLIP score for three colored entities.

**Similarity Score** We present the scores for the dataset with two entities in Table 3. For SD 1.4, without GSNg our method consistently outperforms InitNO, confirming the importance of carefully selecting the diffusion step to perform the IterRef steps. With GSNg, we surpass all competing methods. While we achieve slightly better performance on datasets with two and three entities, Ours+ is marginally lower for datasets that include

---

[2]https://laion.ai/blog/laion-aesthetics/

Table 2: TIAM performance for prompts containing two or three entities, with and without color specifiers. The subscripts refer to CLIP/aesthetic scores. Best values are in bold, with second-best underlined for SD 1.4. For SD 3, only best values are in bold.

| | IterRef | GSNg | Methods | w/o colors | | with colors | |
| | | | | 2 entities | 3 entities | 2 entities | 3 entities |
|---|---|---|---|---|---|---|---|
| SD 1.4 | 0 | ✗ | Stable Diffusion | $45.4_{32.2/5.5}$ | $8.4_{33.5/5.5}$ | $3.9_{34.6/5.4}$ | $0.1_{34.5/5.4}$ |
| | 1 | ✗ | InitNO | $62.1_{33.1/5.5}$ | $14.2_{34.3/5.4}$ | $7.2_{35.7/5.4}$ | $0.2_{35.5/5.3}$ |
| | | | Ours | $65.8_{33.7/5.5}$ | $23.1_{35.4/5.5}$ | $8.7_{36.4/5.4}$ | $0.4_{36.3/5.4}$ |
| | 3 | ✓ | Divide&Bind | $69.9_{33.7/5.5}$ | $33.6_{35.9/5.4}$ | $11.3_{36.1/5.4}$ | $0.5_{36.1/5.3}$ |
| | | | Attend&Excite | $71.4_{34.0/5.5}$ | $32.0_{35.9/5.4}$ | $10.5_{36.9/5.4}$ | $0.6_{36.9/5.3}$ |
| | | | InitNO+ | $75.0_{34.1/5.5}$ | $34.2_{36.0/5.4}$ | $11.9_{37.1/5.4}$ | $1.0_{37.3/5.3}$ |
| | 0 | ✓ | Syngen | $\underline{78.5}_{34.1/5.4}$ | $\underline{39.2}_{36.5/5.4}$ | $\underline{20.4}_{37.1/5.3}$ | $\underline{2.4}_{36.8/5.3}$ |
| | 1 | ✓ | Syngen+ | $75.8_{33.8/5.3}$ | $36.2_{36.2/5.4}$ | $20.1_{37.1/5.3}$ | $1.9_{36.9/5.3}$ |
| | | | Ours+ | $\mathbf{81.1}_{34.2/5.4}$ | $\mathbf{45.8}_{36.7/5.4}$ | $\mathbf{20.5}_{37.1/5.3}$ | $\mathbf{2.8}_{37.1/5.3}$ |
| SD 3 | 0 | ✗ | Stable Diffusion | $82.8_{34.8/5.5}$ | $63.4_{37.9/5.5}$ | $\mathbf{27.3}_{38.2/5.4}$ | $9.69_{39.4/5.3}$ |
| | 1 | ✗ | Ours | $\mathbf{84.5}_{34.9/5.6}$ | $\mathbf{70.7}_{38.2/5.6}$ | $24.2_{38.1/5.4}$ | $\mathbf{9.71}_{39.6/5.4}$ |

Table 3: Similarity scores based on (Chefer et al., 2023) for two entities. Best values are in bold, with second-best underlined for SD 1.4. For SD 3, only best values are in bold.

| | IterRef | GSNg | Methods | Full Prompt | Minimum Object | Text-Text |
|---|---|---|---|---|---|---|
| SD 1.4 | 0 | ✗ | Stable Diffusion | 0.3313 | 0.2400 | 0.7682 |
| | 1 | ✗ | InitNo | 0.3411 | 0.2512 | 0.7901 |
| | | | Ours | 0.3470 | 0.2564 | 0.7979 |
| | 3 | ✓ | Divide&Bind | 0.3468 | 0.2597 | 0.8065 |
| | | | Attend&Excite | 0.3509 | 0.2634 | 0.8032 |
| | | | InitNO+ | $\underline{0.3520}$ | 0.2638 | 0.8076 |
| | 0 | ✓ | Syngen | 0.3518 | $\underline{0.2640}$ | $\underline{0.8122}$ |
| | 1 | ✓ | Ours+ | **0.3522** | **0.2643** | **0.8133** |
| SD 3 | 0 | ✗ | Stable Diffusion | 0.3529 | 0.2616 | 0.8181 |
| | 1 | ✗ | Ours | **0.3535** | **0.2619** | **0.8190** |

color specifications. This may be attributed to the limitations of CLIP-based metrics in capturing precise syntactic relations (Yuksekgonul et al., 2023). Ours outperforms SD 3 on all datasets, with a minor drop in one metric for two colored entities. Results for the other datasets and further discussion on the limits of this score are reported in the Appendix.

**User Study**  We conducted a subjective user study to evaluate human preferences across various methods on SD 1.4, including 37 candidates. For each comparison, we presented images generated by each method using the same randomly selected prompt and seed, with participants asked to choose the best matches or select none if applicable. The study consisted of two phases. In the first phase, we compared InitNO with Ours, followed by a second phase where we evaluated Syngen, InitNO+, and Ours+. As shown in Table 4, our method demonstrates a significant improvement over InitNO in the one-step IterRef setup, further validating the effectiveness of our approach. Additionally, in Table 5 with guidance, participants chose Ours+ more frequently than the others, indicating superior alignment with the text prompts. Further details about the study are provided in the Appendix.

Table 4: User study: methods without GSNg.

| | Ours | InitNO |
|---|---|---|
| Frequency Selection | **43.1%** | 36.9% |

Table 5: User study: methods with GSNg.

| | Ours+ | Syngen | InitNO+ |
|---|---|---|---|
| Frequency Selection | **57.4%** | 51.9% | 43.3% |

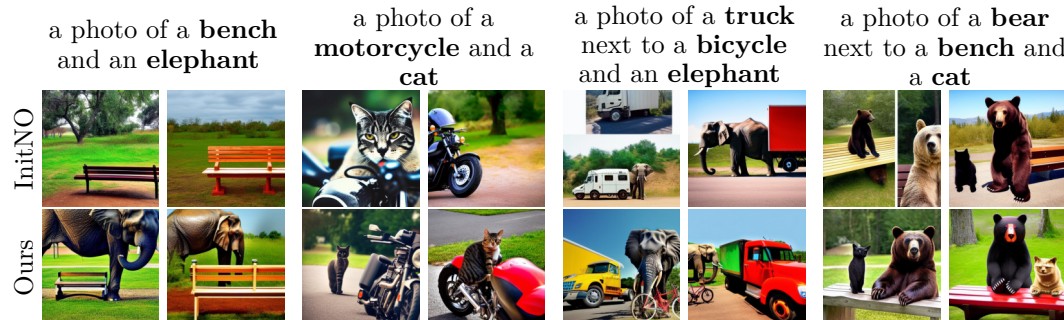

Figure 5: Qualitative comparison between samples generated with methods without GSNg. Images generated with the same set of seeds across the different approaches, using SD 1.4.

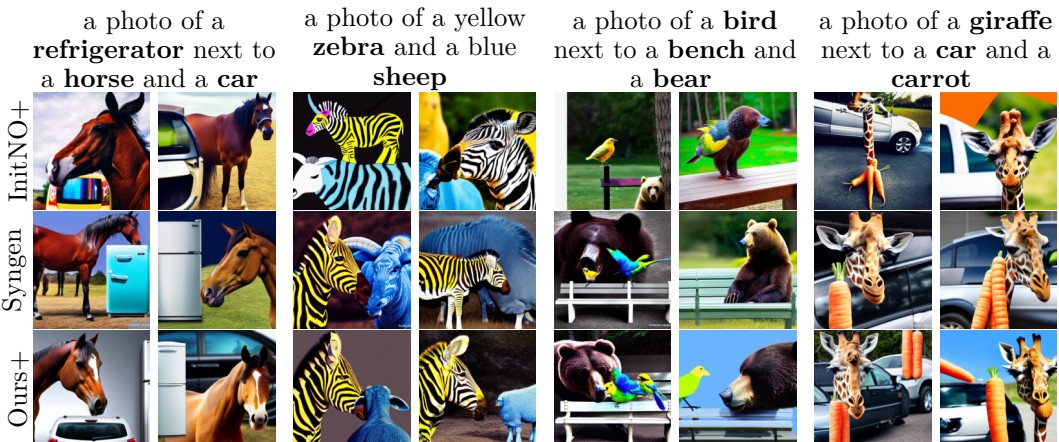

Figure 6: Qualitative comparison between samples generated with methods with GSNg. Images generated with the same set of seeds across the different approaches, using SD 1.4.

### 4.3 Qualitative comparison

We present a qualitative comparison of image generation using the same two seeds with different prompts for methods without GSNg in Figure 5. Our method better mitigates catastrophic neglect *e.g.* InitNO struggles to clearly represent both entities in the prompt *a photo of an elephant and a bench.* Even with challenging prompts containing three entities, our approach yields superior results, as the later IterRef step helps to distinguish the entities more effectively. In Figure 6, we present results for methods employing GSNg. Our method significantly enhances the separation of three objects. For instance, Syngen and InitNO+ fail sometimes to generate certain entities (*e.g.* Syngen: *car* in the first prompt, *bird* in the third prompt; InitNO+: *refrigerator* in the first prompt, *carrot* in the last prompt). Furthermore, our approach better differentiates the entities (*e.g.* Syngen: *sheep* in the second prompt are not distinguishable, while InitNO+ mixes *sheep* with *zebra* in the second image of the second prompt and *bird* with *bear* in the first image of the third prompt). Our approach demonstrates superior performance in effectively generating and distinguishing entities compared to existing approaches. We provide further examples for SD 1.4/3 in the Appendix.

### 4.4 Study of the IterRef placement

We conducted an exhaustive study on the optimal diffusion steps to do the IterRef step (Figure 7). The candidate steps identified in subsection 4.1 align well with the results, as they consistently demonstrate good performance across all datasets. This reinforces the validity of our validation approach for determining IterRef step candidates. We remark that among

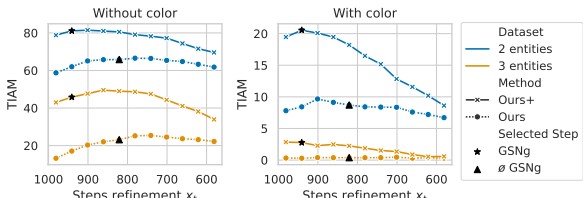

Figure 7: TIAM score according to IterRef step for entities with color (right) and w/o (left).

the configurations tested, optimizing too early was less effective than making adjustments at later stages. However, delaying corrections too much is detrimental, indicating the necessity for a careful trade-off in timing when modifying the signal. We noted that the use of GSNg follows similar trends but consistently yields better results by facilitating slight, continuous adjustments to the signal. We also found that for datasets with color, one can obtain better results by setting different IterRef steps. This conclusion stems from the understanding that color modifications should be implemented early in the diffusion process, as colors appear to be defined at the outset. Making adjustments later may hinder effective integration. In contrast, modifying the signal for entities is more advantageous at later stages, allowing for greater precision in distinguishing between different entities.

## 5 Limitations

The GSN approach is constrained by the model's inherent knowledge, although we can incorporate external information through well-designed GSNs loss. This limitation affects our ability to optimize effectively, as challenges persist *e.g.* rare concept, object confusion, reasoning, counting (Udandarao et al., 2024; Paiss et al., 2023). Consequently, we may encounter failures due to the model's out-of-distribution behavior. Our work has demonstrated that a thorough understanding of signal construction during diffusion allows for the selection of optimization steps that enhance image generation while limiting the number of hyperparameters and the number of IterRef steps, such as optimization thresholds according to the step of diffusion. However, we believe that despite the challenges associated with testing numerous thresholds and hyperparameters, an approach utilizing well-engineered optimization thresholds could improve performance, particularly when considering signal construction. Finally, like other GSN methods, our approach requires back-propagation through the U-net, which is computationally intensive.

## 6 Conclusion

In this study, we improve the application of GSN criteria by exploring how the signal evolves during the diffusion process. We presented a method for identifying and validating an optimal refinement step. Our findings show that while early-stage signal modifications are less effective, timely adjustments can lead to significant performance improvements, enabling the generation of semantically aligned images and achieving state-of-the-art results, as demonstrated through extensive experiments. Furthermore, this approach reduces the number of hyperparameters and IterRef compared to some SOTA methods *e.g.* InitNO, simplifying the model setup and enhancing overall efficiency. We observed that the position of the IterRef step depends on the specific elements we are seeking to correct. For example, color modifications should occur earlier in the process, while adjustments to entities can be made slightly later. Future developments of GSN methods could build on these insights by selecting refinement steps tailored to the particular aspects being adjusted. Additionally, incorporating a reminder loss (Agarwal et al., 2023) could further enhance the approach by providing the model with a memory of the signal across diffusion steps.

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

# A  APPENDIX

The appendix is summarized as follows:

- Section A.1: detailed descriptions of the implementations and methods used,
- Section A.2: detailed description of the use of Stable Diffusion 3 with results on validation datasets,
- Section A.3: an overview of the TIAM evaluation process,
- Section A.4: a summary of the evaluation framework from Attend&Excite, along with additional results,
- Section A.5: detailed information about the user study,
- Section A.6: additional comparative sample outputs,
- Section A.7: values for the figures in the main document and supplementary results, including Section A.7.1 for the validation set and Section A.7.2 for the test set.

## A.1 Text-to-Image Methods Setup For Stable Diffusion 1.4

We provide here some implementation details about Stable Diffusion 1.4. and methods used.

**Stable-Diffusion version 1.4 (SD 1.4)**  We use the model hosted on HuggingFace[3] with the DDPM Scheduler[4] and 50 sampling steps. All the methods were performed with a Classifier Free Guidance (Ho & Salimans, 2022) of 7.5.

**Attend&Excite**  We utilize the implementation provided by the Diffusers library[5]. The iterative refinement occurs at the sampling steps 0, 10, and 20, where the loss must reach specified thresholds of 0.05, 0.5, and 0.8, respectively. A maximum of 20 iterative refinement steps is performed. The learning rate decreases progressively with each sampling step, starting at an initial value of 20. They perform the GSN guidance for the 25 first sampling steps.

**Divide and Bind**  We utilize the official implementation[6]. We follow the authors' recommendation and use the *tv* loss for the prompts without colors and the *tv bind* loss for the prompts with colors. The iterative refinement occurs at the sampling steps 0, 10, and 20, where the losses must reach specified thresholds of 0.05, 0.2, and 0.3, respectively. A maximum of 50 iterative refinement steps is performed. The learning rate decreases progressively with each sampling step, starting at an initial value of 20. They perform the GSN guidance for the 25 first sampling steps.

**InitNO**  We utilize the official implementation provided in the repository[7]. The authors designed a loss function comprising three components: self-attention loss, cross-attention loss, and KL divergence loss. During the multi-round step, an iterative refinement is performed. If the defined thresholds for the cross-attention and self-attention losses are not met, the optimization is repeated by sampling a new starting latent, up to a maximum of five attempts. If the objectives remain unattainable, inference is conducted using the optimized starting latent representation that achieves the best score relative to the objectives. The KL divergence loss is applied exclusively during the boosting step, where optimization is performed after each back-propagation on the attention losses to ensure that the starting latent image remains within an appropriate interval. Iterative refinement steps are also conducted at sampling steps 10 and 20. For both the boosting step and iterative refinement, the losses must meet specified thresholds of 0.2 for the cross-attention loss and 0.3 for the self-attention loss. The learning rate decreases progressively with each sampling step, beginning at an initial value of 20. Additionally, GSN guidance is applied for the first 25 sampling steps.

Additionally, we discovered in the code that the implementation includes a *clean cross-attention loss*, which applies a specialized processing of the attention maps using Otsu thresholding during the multi-round step and GSNg. The code also incorporates a *cross-attention alignment loss* for the GSNg, seemingly designed to encourage consistency in token activation zones across diffusion steps. To the best of our knowledge, these details are not mentioned in the main paper.

**Syngen**  We utilize the official implementation[8]. They apply only a GSN guidance for the first 25 sampling steps. They use a learning rate of 20.

Syngen is designed to accept prompts that consist solely of entities with attributes. For instance, when the prompt is "a photo of a cat and a dog", the cross-attention maps corresponding to "a photo of" are utilized. To enhance the results, we remove the cross-attention maps associated with the initial tokens. This adjustment led to an approximate

---

[3]`https://huggingface.co/CompVis/stable-diffusion-v1-4`

[4]`https://huggingface.co/docs/diffusers/api/schedulers/ddpm`

[5]`https://huggingface.co/docs/diffusers/api/pipelines/attend_and_excite`

[6]`https://github.com/boschresearch/Divide-and-Bind`

[7]`https://github.com/xiefan-guo/initno`

[8]`https://github.com/RoyiRa/Linguistic-Binding-in-Diffusion-Models`

increase of 1 in performance during the experiments. The scores reported for Syngen in the paper reflect these beneficial modifications.

Table 6: TIAM score on the different datasets with Syngen and an iterative refinement step using the Syngen criterion.

| $n$ shift of latent image | Methods | w/o colors | | with colors | |
|---|---|---|---|---|---|
| | | 2 entities | 3 entities | 2 entities | 3 entities |
| 20 | Syngen+ | $77.81_{33.98/5.38}$ | $36.17_{36.32/5.39}$ | $20.08_{37.07/5.3}$ | $1.88_{36.85/5.27}$ |
| 50 | Syngen+ | $75.81_{33.78/5.33}$ | $36.23_{36.17/5.35}$ | $18.23_{37.06/5.26}$ | $1.9_{36.97/5.25}$ |

We attempted to introduce one refinement step for Syngen. Specifically, we applied a refinement step at the first sampling step, similar to InitNO, and conducted 20 and 50 optimization iterations using the loss function of Syngen. The Adam optimizer was employed with a learning rate of $1 \times 10^{-2}$. The TIAM scores are reported in Table 6. However, we did not achieve better results compared to configurations without refinement steps. While improvements may be possible, further research is required to identify optimal hyperparameters.

**Ours**  Following the Attend&Excite framework, we apply Gaussian smoothing to the attention maps using a kernel size of 3 and a standard deviation of 0.5. During the iterative refinement step, we conduct 50 latent image shifts without aiming to achieve a specific threshold. For the configuration utilizing GSN guidance, we incorporate the Syngen GSN guidance after proceeding with the iterative refinement step.

## A.2 STABLE DIFFUSION 3

We use the implementation available on Hugging Face[9] with the default scheduler, Flow-MatchEulerDiscreteScheduler[10], configured with 28 sampling steps, a Classifier-Free Guidance (Ho & Salimans, 2022) of 7.0, and bfloat16 precision for image generation. For IterRef, we apply an Adam optimizer with a learning rate of $1 \times 10^{-2}$ and 50 steps of optimization.

Stable Diffusion 3 (SD 3) is a Flow Matching model designed to construct a probabilistic path between two distributions, $p_0$ and $p_1$, where $p_0$ is the target distribution and $p_1 \sim \mathcal{N}(0, I)$. The model learns to transport points from one distribution to another. The latent image transport path can be interpreted as a denoising process, with noise progressively removed in a manner analogous to image destruction. Specifically, the latent image $x_t$ is sampled using the reparameterization trick, involving the interpolation of the image and noise. As demonstrated by Rissanen et al. (2023), isotropic noise suppresses frequency components in the data that have a lower power spectral density than the variance of the noise. Consequently, the model initially reconstructs lower frequencies and subsequently refines higher frequencies, similar to the process observed in diffusion models. During denoising, the signal can be refined to ensure alignment with the desired output using the GSN approach. Additionally, our method can be applied to select the optimal step in the denoising process. While feature extraction in Stable Diffusion models 1.4 and 1.5 is well-documented, to the best of our knowledge, this has not been extensively explored for Stable Diffusion 3, which uses a transformer-based architecture. In this architecture, T5 and CLIP serve as two distinct encoders for guidance. The model incorporates two independent transformers, each operating within its own modality space (image patches and text), while taking the other modality into account when processing the attention. We first describe how we process and extract attention maps and secondly, how we select a potential nice step to refine the latent image.

**Extraction of the attention maps** Stable Diffusion 3 consists of 24 transformer blocks. The latent image, represented as $x_t \in \mathbb{R}^{H \times W \times c}$, where $c$ is the number of channels in the latent space, and $H, W$ are the height and width, is patchified to produce a sequence of tokens $z \in \mathbb{R}^{hw \times d}$, where $hw = \frac{1}{2}H \times \frac{1}{2}W$, and $d$ is the token embedding dimension.

The textual embedding $t$ is formed by concatenating the embeddings from CLIP and T5 and projecting them into the same dimension $d$. This results in $t \in \mathbb{R}^{(n_{\text{CLIP}} + n_{\text{T5}}) \times d}$, where $n_{\text{CLIP}}$ and $n_{\text{T5}}$ represent the number of tokens from CLIP and T5, respectively.

When processing attention, the resulting attention maps $A$ are of size $A \in \mathbb{R}^{(hw + n_{\text{CLIP}} + n_{\text{T5}})^2 \times n_{\text{head}}}$, where $n_{\text{head}}$ is the number of attention heads. We extract the attention maps and focus on the subset where the image patches serve as the queries, and the text embeddings act as the keys. This subset is crucial as it captures the relationship between the image latent and textual concepts, ensuring the signal within the latent image aligns with the semantic meaning of the tokens.

To simplify the attention maps, we average across the attention heads and transformer blocks, yielding $A \in \mathbb{R}^{hw \times (n_{\text{CLIP}} + n_{\text{T5}})}$. We further refine these maps by excluding the special tokens (e.g., the start and end tokens) for both CLIP and T5, as these tend to dominate the attention distribution without contributing meaningful semantic information. The attention maps are reweighted using a softmax operation and Gaussian smoothing, as proposed by Chefer et al. (2023) for Stable Diffusion 1.4. For subject tokens that span multiple tokens (e.g., due to subword tokenization), we average their respective attention maps. Finally, the attention maps from CLIP and T5 are aligned and combined by averaging, producing the final attention maps, $A \in \mathbb{R}^{hw \times \mathcal{S}}$, used to guide the latent space adjustment. The loss function described in the main paper is applied to modify the latent representation accordingly. However, further investigation is required to determine whether extracting attention maps from all transformer blocks is necessary. Preliminary observations suggest that the first and last transformer blocks lack clear semantic correspondence with spatial features, as revealed through visualizations. A selective approach to choosing transformer

---

[9]https://huggingface.co/stabilityai/stable-diffusion-3-medium-diffusers

[10]https://huggingface.co/docs/diffusers/api/schedulers/flow_match_euler_discrete

 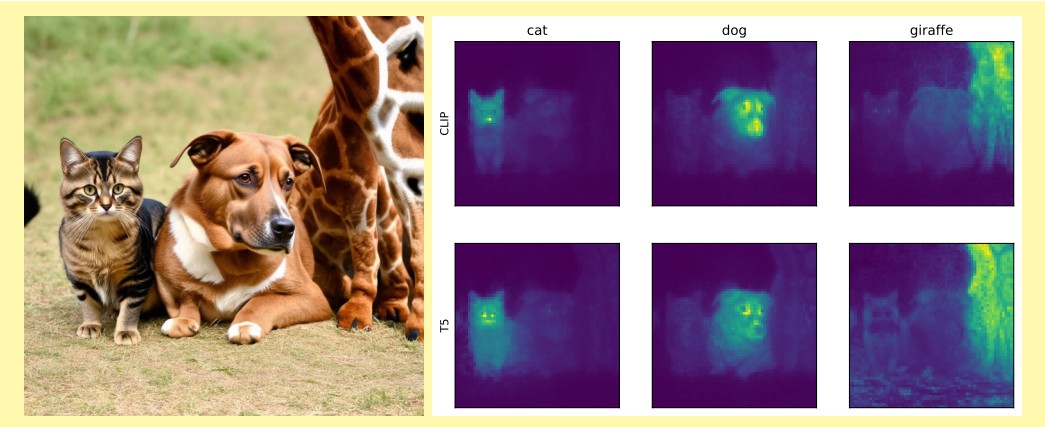

Figure 8: On the left, the image generated by Stable Diffusion 3 for the prompt "a photo of a cat next to a dog and a giraffe". On the right, extracted attention maps for CLIP and T5 tokens, averaged across all diffusion steps and transformer blocks. For words represented by multiple tokens, the attention maps are further averaged.

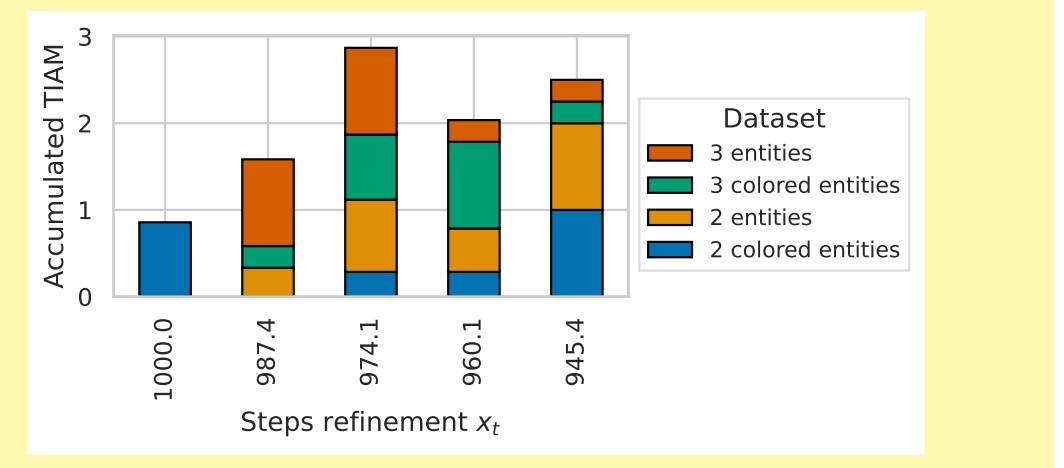

Figure 9: Accumulated TIAM scores for Stable Diffusion 3.

blocks, based on a detailed analysis, could lead to more effective results. This refined attention extraction process may also serve as a foundation for future work in developing semantic map extraction techniques such as Tang et al. (2023). We include an example of the extracted attention maps without any processing, averaged across blocks and generation steps in the Figure 8. The semantic correspondence between text representations and visual representations can be observed.

**Optimal IterRef step selection** We evaluate the first 5 sampling steps. For each validation dataset, we generate 8 images per prompt using the same 8 seeds and compute the TIAM score. The scores are standardized using a min-max scaler for each dataset, and the *accumulated standardized TIAM* across the IterRef steps is shown in Figure 9. The third sampling step is optimal as it performs consistently well across all datasets. The non-standardized values are presented in Figure 9.

Table 7: TIAM score for Stable Diffusion 3 according to the steps used for the refinement with datasets with 10 prompts (when color TIAM score ground truth colors that is displayed).

| step of iterative refinement | 2 entities | | 3 entities | |
|---|---|---|---|---|
| | wo color | color | wo color | color |
| 945.4 | 88.75 | 28.75 | 62.50 | 13.75 |
| 960.1 | 85.00 | 22.50 | 62.50 | 17.50 |
| 974.1 | 87.50 | 22.50 | 66.25 | 16.25 |
| 987.4 | 83.75 | 20.00 | 66.25 | 13.75 |
| 1000 | 81.25 | 27.50 | 61.25 | 12.50 |

## A.3   TIAM

We use the following set of 24 COCO labels $O = $ *bicycle, car, motorcycle, truck, donut, bench, bird, cat, dog, horse, sheep, cow, elephant, bear, zebra, giraffe, banana, apple, broccoli, carrot, chair, couch, oven, refrigerator.* The templates are :

- two entities: "a photo of $det(o_1)$ $o_1$ and $det(o_2)$ $o_2$"
- three entities: "a photo of $det(o_1)$ $o_1$" next to $det(o_2)$ $o_2$ and $det(o_3)$ $o_3$"

with $o_i \in O$ and $det(.)$ the correct article depending on the object $o_i$.

With attribute, we retake the same set of objects $O$ with the following set of attributes $\mathcal{A} = \{red, green, blue, purple, pink, yellow\}$. We used the following templates:

- two colored entities : "a photo of $det(a_1)$ $a_1$ $o_1$ and $det(a_2)$ $a_2$ $o_2$ "
- three colored entities : "a photo of $det(a_1)$ $a_1$ $o_1$, $det(a_2)$ $a_2$ $o_2$ and $det(a_3)$ $a_3$ $o_3$"

with $o_i \in O$, $a_i \in \mathcal{A}$ and $det(.)$ the correct article depending on the attribute $a_i$.

We then generate all the combinations and following Grimal et al. (2024), we can obtain an approximation by sampling 300 prompts and generate 16 images per prompt using the same seeds. We follow the main implementation, we detect the presence of an object with YOLOv8 (Jocher et al., 2023) and accept the presence of the object if confidence $\geq 0.25$. For an image to be considered well-generated, the requested entities must be correctly detected. Additionally, in the case of colored entities, both the entity detection and the color attribution must be accurate.

In comparison with the Attend&Excite evaluation setup, in the case of attribute binding, each entity is qualified by an attribute. In Table 8 and Table 9, we present the distribution of couple and trio of meta-class of entities following this classification of different labels :

- Animal : bird, cat, dog, horse, sheep, cow, elephant, bear, zebra, giraffe.
- Objects : bicycle, car, motorcycle, truck, donut, bench, banana, apple, broccoli, carrot, chair, couch, oven, refrigerator.

For reproduction of the experiments, we release the datasets[11].

Table 8: Number of associations of classes in the datasets of prompts with two entities.

| Dataset | Animal-Animal | Animal-Object | Object-Object |
|---|---|---|---|
| 2 entities | 47 | 151 | 102 |
| 2 colored entities | 45 | 140 | 115 |
| 2 entities + 2 colored entities | 92 | 291 | 217 |

Table 9: Number of associations of classes in the datasets of prompts with three entities.

| Dataset | Animal-Animal Animal | Animal-Animal Object | Animal-Object Object | Object-Object Object |
|---|---|---|---|---|
| 3 entities | 17 | 98 | 135 | 50 |
| 3 colored entities | 12 | 87 | 139 | 62 |
| 3 entities + 3 colored entities | 29 | 185 | 274 | 112 |

---

[11] https://huggingface.co/datasets/anonymous4review

## A.4 Attend&Excite evaluation

Attend&Excite uses CLIP[12] (Radford et al., 2021) and BLIP[13] (Li et al., 2022) for evaluation. They compute scores using the cosine similarity of CLIP embedding. To have an average semantic embedding to compute, they create 80 derived of the prompt using 80 templates such as

```
"a bad photo of a {}", "a photo of many {}", "a sculpture of a {}"
```

available on their github[14]. Then they fill out the {} with the entities in the original prompt. After that, they compute the CLIP embedding and average among the 80 created prompts.

We detail how they compute each score:

- *Full Prompt Similarity*: Cosine similarity between the CLIP embedding of the generated image and the average embedding from the 80 templates.

- *Minimum Object Similarity*: Average text CLIP embedding for each entity is computed from the templates. Cosine similarity between the generated image and each average embedding corresponding to an entity and the minimum similarity is reported.

- *Text-Text Similarity*: The caption of the generated image (with BLIP) is compared with the average embedding of the 80 templates of the original prompt using cosine similarity.

Table 10: Similarity scores based on (Chefer et al., 2023) for two entities. The exponents present the standard deviations. Best values are in bold, with second-best underlined for SD 1.4. For SD 3, only best values are in bold.

| | IterRef | GSNg | Methods | Full Prompt | Minimum Object | Text-Text |
|---|---|---|---|---|---|---|
| **SD 1.4** | 0 | ✗ | Stable Diffusion | $0.3313^{\pm 0.0375}$ | $0.2400^{\pm 0.0377}$ | $0.7682^{\pm 0.1017}$ |
| | 1 | ✗ | InitNo | $0.3411^{\pm 0.0350}$ | $0.2512^{\pm 0.0328}$ | $0.7901^{\pm 0.1012}$ |
| | | | Ours | $0.3470^{\pm 0.0336}$ | $0.2564^{\pm 0.0308}$ | $0.7979^{\pm 0.0990}$ |
| | 3 | ✓ | Divide&Bind | $0.3468^{\pm 0.0295}$ | $0.2597^{\pm 0.0246}$ | $0.8065^{\pm 0.0962}$ |
| | | | Attend&Excite | $0.3509^{\pm 0.0296}$ | $0.2634^{\pm 0.0226}$ | $0.8032^{\pm 0.0964}$ |
| | | | InitNO+ | $\underline{0.3520}^{\pm 0.0285}$ | $0.2638^{\pm 0.0211}$ | $0.8076^{\pm 0.0951}$ |
| | 0 | ✓ | Syngen | $0.3518^{\pm 0.0282}$ | $\underline{0.2640}^{\pm 0.0231}$ | $\underline{0.8122}^{\pm 0.0970}$ |
| | 1 | ✓ | Ours+ | $\mathbf{0.3522}^{\pm 0.0270}$ | $\mathbf{0.2643}^{\pm 0.0213}$ | $\mathbf{0.8133}^{\pm 0.0960}$ |
| **SD 3** | 0 | ✗ | Stable Diffusion | $0.3529^{\pm 0.0294}$ | $0.2616^{\pm 0.0244}$ | $0.8181^{\pm 0.0921}$ |
| | 1 | ✗ | Ours | $\mathbf{0.3535}^{\pm 0.0281}$ | $\mathbf{0.2619}^{\pm 0.0226}$ | $\mathbf{0.8190}^{\pm 0.0928}$ |

In our case, we compute the score for each dataset. In addition to the results presented in the main paper, we provide average evaluations for the all datasets with the standard deviation:

- two entities Table 10,

- three entities Table 11,

- two colored entities Table 12,

- three colored entities Table 13.

---

[12]https://huggingface.co/openai/clip-vit-base-patch16

[13]https://huggingface.co/Salesforce/blip-image-captioning-base

[14]https://github.com/yuval-alaluf/Attend-and-Excite/

Table 11: Similarity scores based on (Chefer et al., 2023) for three entities. The exponents present the standard deviations. Best values are in bold, with second-best underlined for SD 1.4. For SD 3, only best values are in bold.

| | IterRef | GSNg | Methods | Full Prompt | Minimum Object | Text-Text |
|---|---|---|---|---|---|---|
| **SD 1.4** | 0 | ✗ | Stable Diffusion | $0.3450^{\pm 0.0381}$ | $0.2063^{\pm 0.0293}$ | $0.7322^{\pm 0.1012}$ |
| | 1 | ✗ | InitNo | $0.3528^{\pm 0.0364}$ | $0.2106^{\pm 0.0302}$ | $0.7408^{\pm 0.1041}$ |
| | | | Ours | $0.3639^{\pm 0.0356}$ | $0.2204^{\pm 0.0305}$ | $0.7568^{\pm 0.1013}$ |
| | 3 | ✓ | Divide&Bind | $0.3687^{\pm 0.0341}$ | $0.2282^{\pm 0.0281}$ | $0.7618^{\pm 0.1038}$ |
| | | | Attend&Excite | $0.3708^{\pm 0.0326}$ | $0.2327^{\pm 0.0252}$ | $0.7582^{\pm 0.1026}$ |
| | | | InitNO+ | $0.3719^{\pm 0.0324}$ | $\underline{0.2331}^{\pm 0.0240}$ | $0.7594^{\pm 0.1048}$ |
| | 0 | ✓ | Syngen | $\underline{0.3750}^{\pm 0.0311}$ | $0.2320^{\pm 0.0277}$ | $\underline{0.7660}^{\pm 0.1066}$ |
| | 1 | ✓ | Ours+ | $\mathbf{0.3772}^{\pm 0.0299}$ | $\mathbf{0.2349}^{\pm 0.0253}$ | $\mathbf{0.7698}^{\pm 0.1056}$ |
| **SD 3** | 0 | ✗ | Stable Diffusion | $0.3833^{\pm 0.0309}$ | $0.2346^{\pm 0.0255}$ | $0.7876^{\pm 0.0966}$ |
| | 1 | ✗ | Ours | $\mathbf{0.3863}^{\pm 0.0281}$ | $\mathbf{0.2373}^{\pm 0.0226}$ | $\mathbf{0.7908}^{\pm 0.0951}$ |

Table 12: Similarity scores based on (Chefer et al., 2023) for two colored entities. The exponents present the standard deviations. Best values are in bold, with second-best underlined for Stable Diffusion 1.4. For Stable Diffusion 3, only best values are in bold.

| | IterRef | GSNg | Methods | Full Prompt | Minimum Object | Text-Text |
|---|---|---|---|---|---|---|
| **SD 1.4** | 0 | ✗ | Stable Diffusion | $0.3527^{\pm 0.0343}$ | $0.2483^{\pm 0.0393}$ | $0.7208^{\pm 0.1130}$ |
| | 1 | ✗ | InitNo | $0.3639^{\pm 0.0337}$ | $0.2618^{\pm 0.0363}$ | $0.7329^{\pm 0.1120}$ |
| | | | Ours | $0.3720^{\pm 0.0330}$ | $0.2699^{\pm 0.0329}$ | $0.7420^{\pm 0.1143}$ |
| | 3 | ✓ | Divide&Bind | $0.3688^{\pm 0.0303}$ | $0.2711^{\pm 0.0297}$ | $0.7317^{\pm 0.1180}$ |
| | | | Attend&Excite | $0.3767^{\pm 0.0298}$ | $0.2782^{\pm 0.0267}$ | $0.7422^{\pm 0.1150}$ |
| | | | InitNO+ | $\mathbf{0.3787}^{\pm 0.0289}$ | $\mathbf{0.2792}^{\pm 0.0256}$ | $0.7453^{\pm 0.1129}$ |
| | 0 | ✓ | Syngen | $\underline{0.3784}^{\pm 0.0309}$ | $0.2774^{\pm 0.0296}$ | $\mathbf{0.7534}^{\pm 0.1175}$ |
| | 1 | ✓ | Ours+ | $0.3780^{\pm 0.0304}$ | $\underline{0.2784}^{\pm 0.0280}$ | $\underline{0.7483}^{\pm 0.1196}$ |
| **SD 3** | 0 | ✗ | Stable Diffusion | $0.3863^{\pm 0.0273}$ | $0.2806^{\pm 0.0259}$ | $\mathbf{0.7731}^{\pm 0.1225}$ |
| | 1 | ✗ | Ours | $\mathbf{0.3864}^{\pm 0.0262}$ | $\mathbf{0.2812}^{\pm 0.0241}$ | $0.7708^{\pm 0.1238}$ |

In the context of one-step refinement, our method consistently outperforms InitNO. With GSN guidance, we observe slight improvements for the three-entities datasets compared to other approaches; however, our scores are lower for datasets that include colors, which may be explained by the limitations of CLIP-based metrics, as they have a bags-of-words behavior(Yuksekgonul et al., 2023): inadequate relational understanding, frequent errors in associating objects with their attributes, and a significant lack of sensitivity to the order of elements. In addition, the close similarity of the scores, along with the large standard deviations, suggests that this evaluation used might not be accurately detecting significant differences between methods. This brings into question whether the results are truly meaningful, highlighting the need for further research to assess the validity and reliability of this metric in evaluating text-image alignment performance.

Table 13: Similarity scores based on (Chefer et al., 2023) for three colored entities. The exponents present the standard deviations. Best values are in bold, with second-best underlined for Stable Diffusion 1.4. For Stable Diffusion 3, only best values are in bold.

| | IterRef | GSNg | Methods | Full Prompt | Minimum Object | Text-Text |
|---|---|---|---|---|---|---|
| **SD 1.4** | 0 | ✗ | Stable Diffusion | $0.3519^{\pm 0.0331}$ | $0.2148^{\pm 0.0297}$ | $0.6505^{\pm 0.1017}$ |
| | 1 | ✗ | InitNo | $0.3633^{\pm 0.0317}$ | $0.2211^{\pm 0.0305}$ | $0.6578^{\pm 0.1026}$ |
| | | | Ours | $0.3707^{\pm 0.0313}$ | $0.2274^{\pm 0.0299}$ | $0.6621^{\pm 0.1043}$ |
| | 3 | ✓ | Divide&Bind | $0.3689^{\pm 0.0298}$ | $0.2305^{\pm 0.0279}$ | $0.6542^{\pm 0.1016}$ |
| | | | Attend&Excite | $0.3772^{\pm 0.0292}$ | $\underline{0.2388}^{\pm 0.0261}$ | $0.6557^{\pm 0.1024}$ |
| | | | InitNO+ | $\mathbf{0.3809}^{\pm 0.0297}$ | $\mathbf{0.2403}^{\pm 0.0256}$ | $0.6565^{\pm 0.1048}$ |
| | 0 | ✓ | Syngen | $0.3754^{\pm 0.0305}$ | $0.2308^{\pm 0.0294}$ | $\mathbf{0.6715}^{\pm 0.1065}$ |
| | 1 | ✓ | Ours+ | $\underline{0.3776}^{\pm 0.0302}$ | $0.2346^{\pm 0.0290}$ | $\underline{0.6673}^{\pm 0.1065}$ |
| **SD 3** | 0 | ✗ | Stable Diffusion | $0.3998^{\pm 0.0253}$ | $0.2460^{\pm 0.0231}$ | $0.6726^{\pm 0.1104}$ |
| | 1 | ✗ | Ours | $\mathbf{0.4025}^{\pm 0.0242}$ | $\mathbf{0.2486}^{\pm 0.0211}$ | $\mathbf{0.6744}^{\pm 0.1104}$ |

## A.5 User study

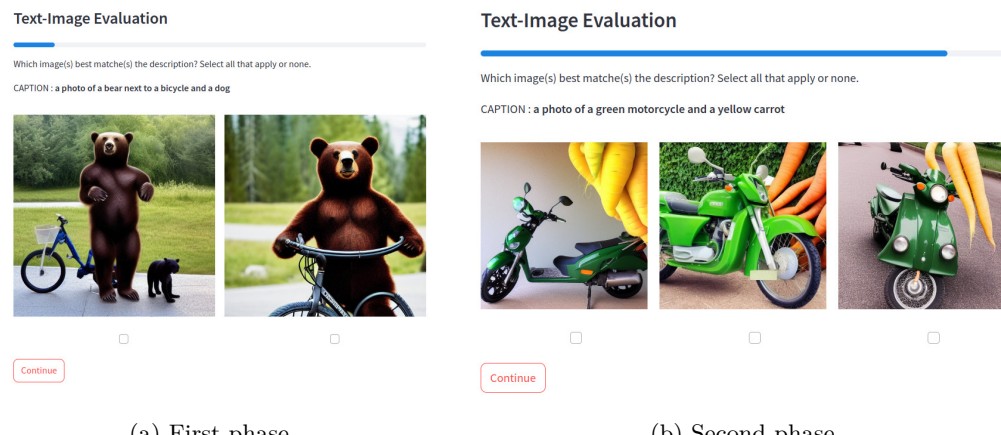

(a) First phase        (b) Second phase

Figure 10: Screenshots of the user study interface.

To compare the methods, we conducted a user study in which participants were shown images generated using the same seed and prompt. The images were randomly sampled from the generated test set. Note that the InitNO method may resample new seeds due to its multi-round iterative refinement step.

Participants were asked to select images that best matched the given prompt. They could choose one, multiple, or none of the images. The study consisted of two phases:

- The first phase involved presenting images from Ours and InitNO, representing methods without GSN guidance. Images were shown from the two entities and three entities datasets.
- The second phase involved presenting images from Ours+, InitNO+, and Syngen, representing methods with GSN guidance. Images were shown from the two entities, two colored entities, and three entities datasets.

The selection of the presented datasets is based on the TIAM score. Without guidance, methods perform too poorly on the two and three colored datasets. With guidance, methods still perform poorly on the three colored dataset.

Each participant was asked to respond to 16 prompts in the first phase and 21 prompts in the second phase. The results from 22 participants, who were shown the same set of images, were used to compute inter-rater reliability using Fleiss' kappa (Fleiss et al., 1971), where 0.5 indicates fair agreement (Landis & Koch, 1977). Figure 10 shows the interface used by participants to select the images.

In total, we had 37 participants, of whom 7 were experts in computer vision. The distribution of participants' age categories is shown in Figure 11.

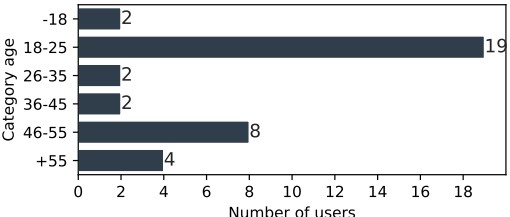

Figure 11: Distribution of participants' ages in the study.

## A.6 MORE QUALITATIVE SAMPLES

We provide more examples of generated images with Stable Diffusion 1.4:

- without GSN guidance in Figure 12, Figure 13, Figure 14, Figure 15, Figure 16,
- with GSN guidance in Figure 17, Figure 18, Figure 19, Figure 20, Figure 21.

We provide examples of generated images with Stable Diffusion 3 in Figure 22, Figure 23, Figure 24, Figure 25 and Figure 26.

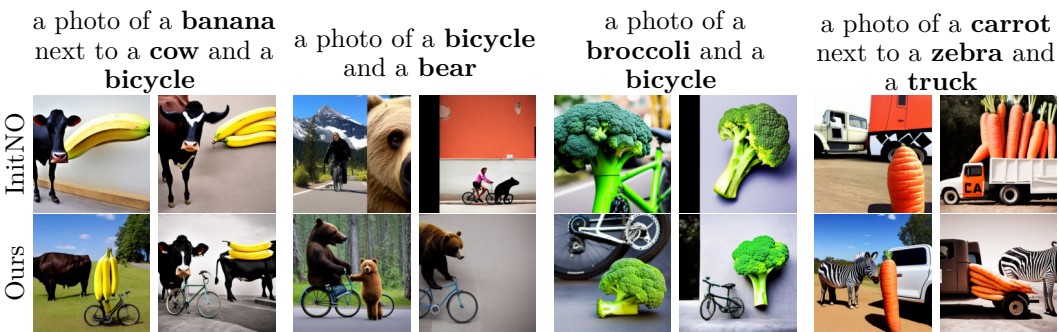

Figure 12: Qualitative comparison between samples generated with methods without GSNg. Images generated with the same set of seeds across the different approaches, using SD 1.4.

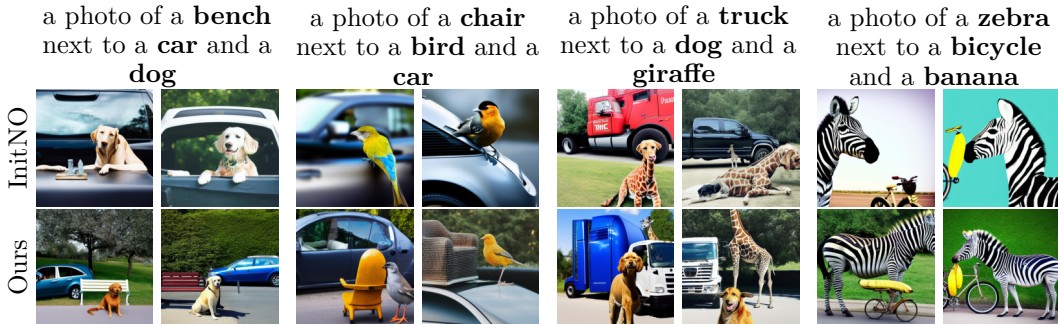

Figure 13: Qualitative comparison between samples generated with methods without GSNg. Images generated with the same set of seeds across the different approaches, using SD 1.4.

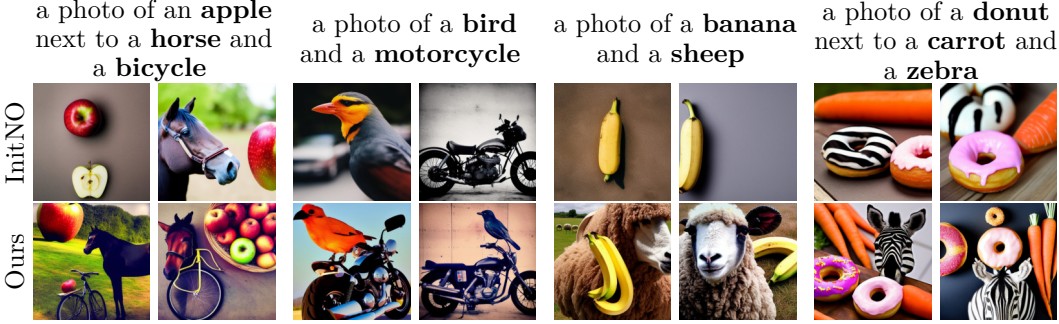

Figure 14: Qualitative comparison between samples generated with methods without GSNg. Images generated with the same set of seeds across the different approaches, using SD 1.4.

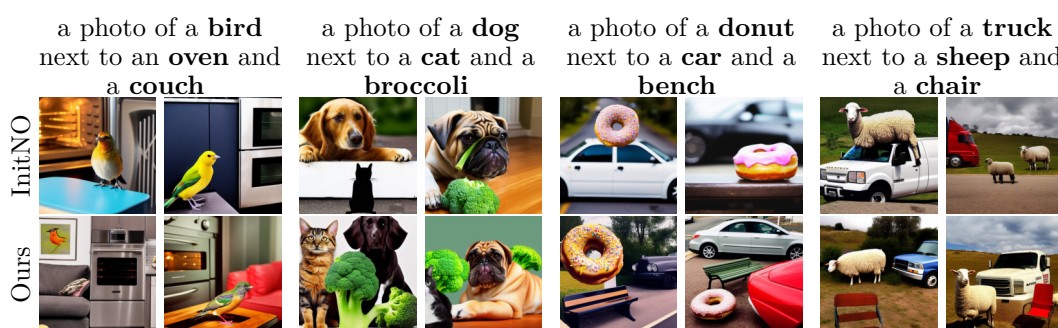

Figure 15: Qualitative comparison between samples generated with methods without GSNg. Images generated with the same set of seeds across the different approaches, using SD 1.4.

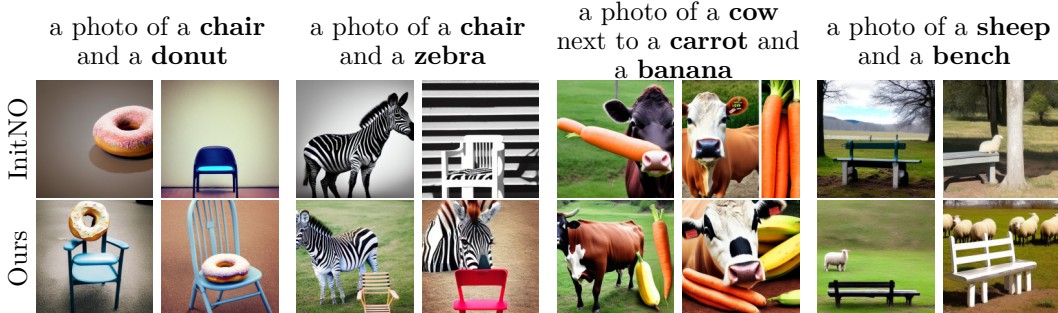

Figure 16: Qualitative comparison between samples generated with methods without GSNg. Images generated with the same set of seeds across the different approaches, using SD 1.4.

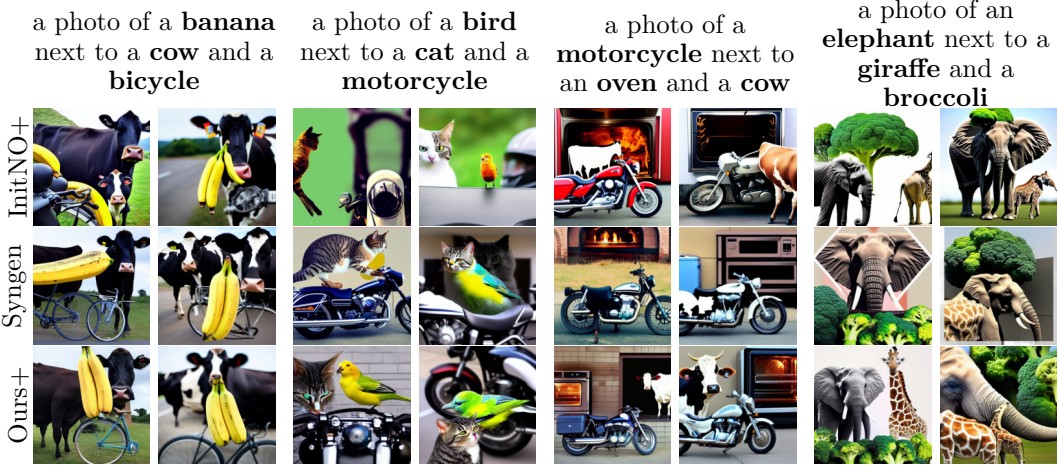

Figure 17: Qualitative comparison between samples generated with methods with GSNg. Images generated with the same set of seeds across the different approaches, using SD 1.4.

a photo of a pink **bird** and a purple **dog**      a photo of a zebra next to a giraffe and a bicycle      a photo of a purple **dog** and a blue **cow**      a photo of a **bird** next to a **truck** and a **motorcycle**

InitNO+    Syngen    Ours+

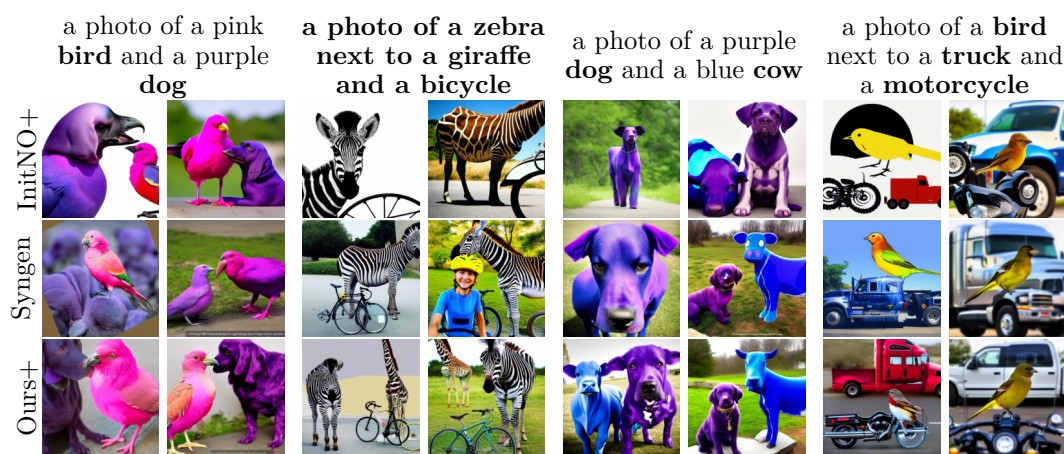

Figure 18: Qualitative comparison between samples generated with methods with GSNg. Images generated with the same set of seeds across the different approaches, using SD 1.4.

a photo of a **dog** next to a **cow** and a **banana**      a photo of a green **couch** and a purple **motorcycle**      a photo of an **apple** next to a **horse** and a **banana**      a photo of a yellow **couch** and a pink **horse**

InitNO+    Syngen    Ours

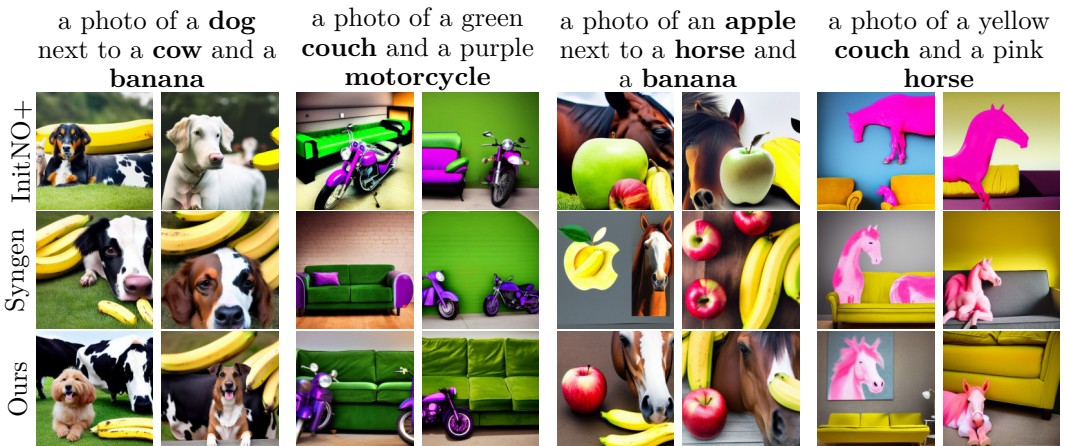

Figure 19: Qualitative comparison between samples generated with methods with GSNg. Images generated with the same set of seeds across the different approaches, using SD 1.4.

a photo of a **chair** and a **bicycle**      a photo of a **couch** next to an **oven** and a **dog**      a photo of a pink **cow** and a red **refrigerator**      a photo of a **horse** next to a **banana** and a **truck**

InitNO+    Syngen    Ours+

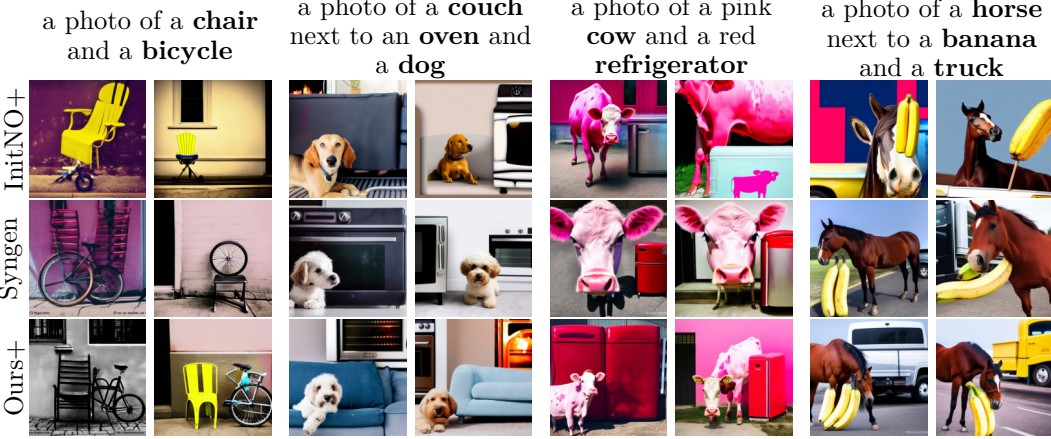

Figure 20: Qualitative comparison between samples generated with methods with GSNg. Images generated with the same set of seeds across the different approaches, using SD 1.4.

a photo of a yellow **zebra** and a red **chair**     a photo of an **elephant** and a **bear**     a photo of a purple **sheep** and a green **zebra**     a photo of a **bird** next to a **bear** and a **donut**

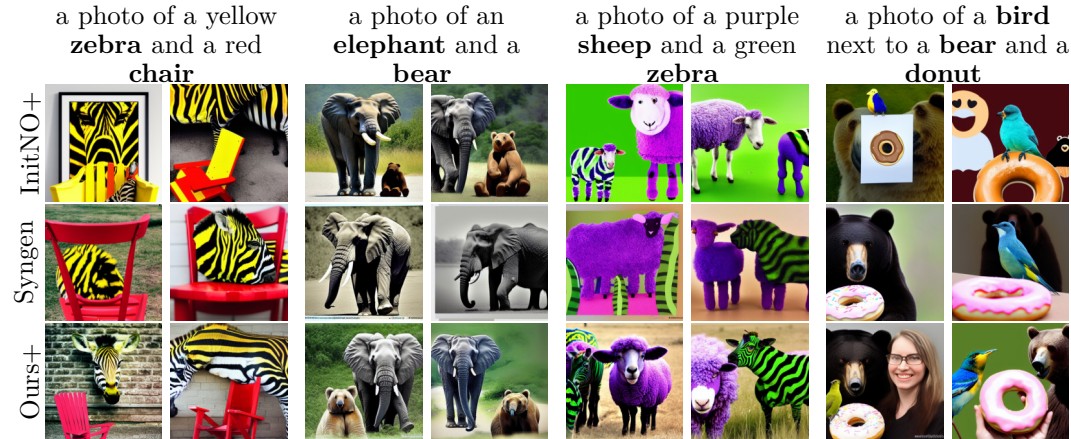

Figure 21: Qualitative comparison between samples generated with methods with GSNg. Images generated with the same set of seeds across the different approaches, using SD 1.4.

a photo of a **bicycle** next to an **elephant** and an **oven**     a photo of a pink **bird** and a blue **cat**     a photo of a **zebra** next to an **oven** and a **horse**     a photo of a **bicycle** next to a **horse** and a **giraffe**

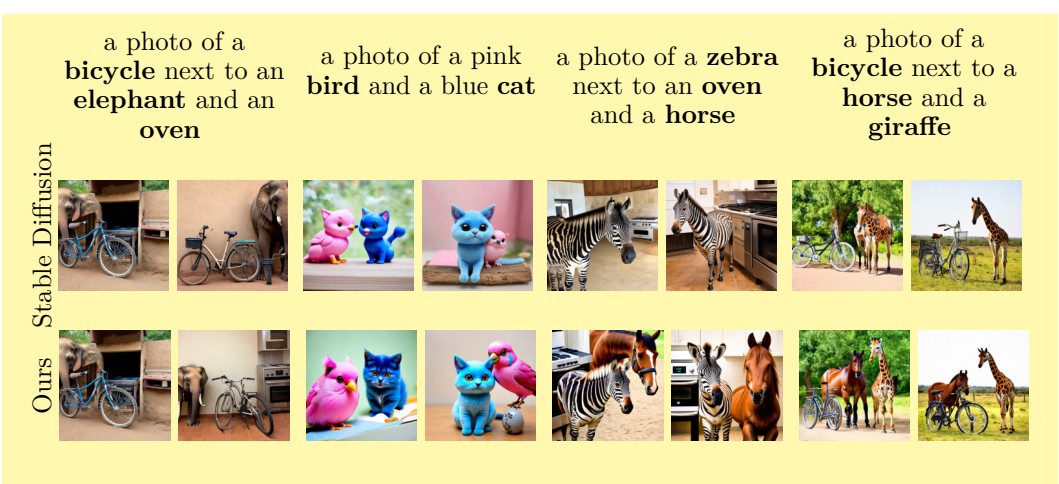

Figure 22: Qualitative comparison between samples generated with SD 3. Images generated with the same set of seeds across the different approaches.

a photo of a **bird** next to a **cat** and a **motorcycle**     a photo of a **car** next to an **elephant** and a **refrigerator**     a photo of a green **zebra** and a purple **cat**     a photo of an **apple** next to a **horse** and a **bicycle**

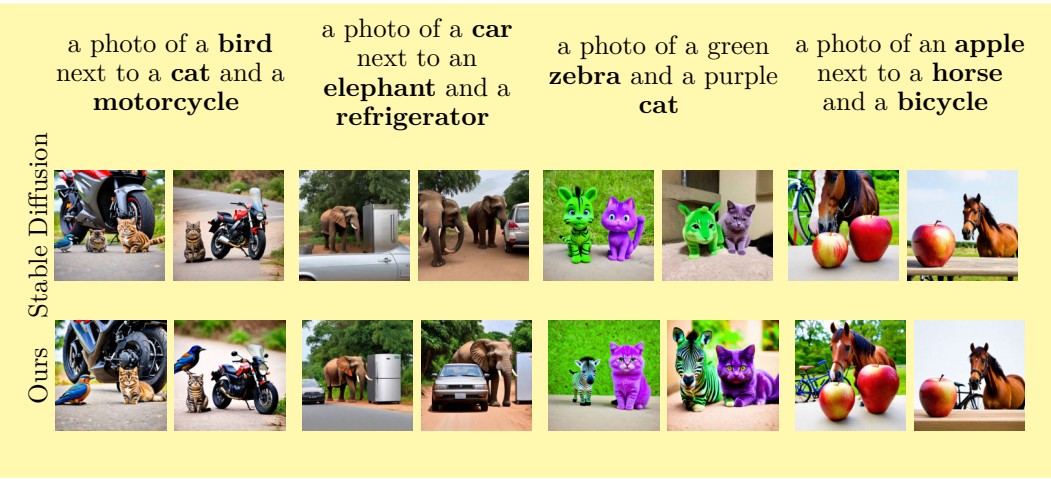

Figure 23: Qualitative comparison between samples generated with SD 3. Images generated with the same set of seeds across the different approaches.

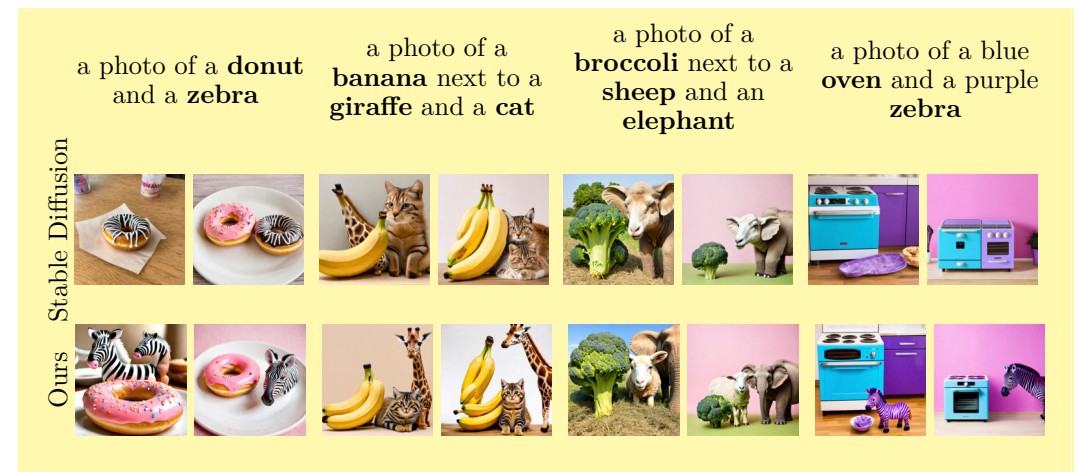

Figure 24: Qualitative comparison between samples generated with SD 3. Images generated with the same set of seeds across the different approaches.

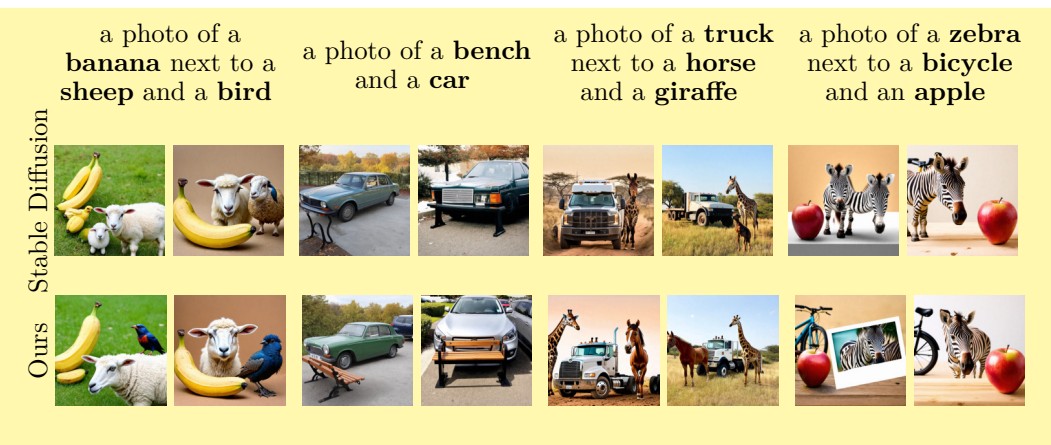

Figure 25: Qualitative comparison between samples generated with SD 3. Images generated with the same set of seeds across the different approaches.

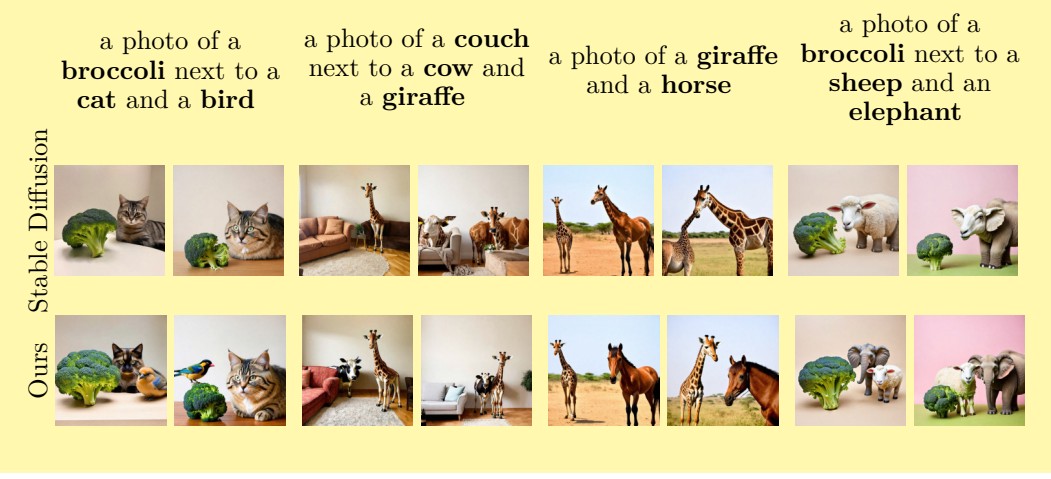

Figure 26: Qualitative comparison between samples generated with SD 3. Images generated with the same set of seeds across the different approaches.

## A.7 Reporting the scores values and additional results

### A.7.1 Validation set

We report the TIAM scores on the validation dataset in Table 14 and we represent the scores as a function of refinement steps used in Figure 27. Additionally, we present the CLIP Score (in Figure 28) and Aesthetic score (Figure 29) according to the refinements steps used.

Table 14: TIAM score according to the steps used for the refinement with datasets with 10 prompts (when color TIAM score ground truth colors that is displayed).

| step of iterative refinement | 2 entities | | | | 3 entities | | | |
| | wo color | | color | | wo color | | color | |
| | ø | GSNg | ø | GSNg | ø | GSNg | ø | GSNg |
| --- | --- | --- | --- | --- | --- | --- | --- | --- |
| 981 | 44.38 | 69.38 | 7.50 | 20.00 | 5.00 | 32.50 | 0.00 | 2.50 |
| 941 | 53.12 | 73.12 | 8.75 | 18.75 | 4.38 | 35.62 | 0.00 | 3.75 |
| 901 | 55.00 | 76.88 | 14.38 | 21.88 | 6.25 | 36.88 | 0.00 | 1.88 |
| 861 | 56.88 | 78.13 | 15.62 | 17.50 | 11.25 | 34.38 | 0.00 | 0.62 |
| 821 | 56.88 | 74.38 | 18.75 | 16.88 | 11.88 | 38.12 | 0.00 | 0.62 |
| 781 | 60.00 | 76.25 | 14.38 | 17.50 | 10.63 | 33.13 | 0.00 | 1.25 |
| 741 | 58.13 | 75.00 | 18.13 | 16.25 | 9.38 | 40.63 | 0.00 | 0.62 |
| 701 | 61.25 | 73.75 | 15.00 | 15.00 | 11.25 | 29.38 | 0.00 | 1.88 |
| 661 | 57.50 | 70.62 | 14.38 | 13.75 | 14.38 | 24.38 | 0.62 | 1.25 |
| 621 | 59.38 | 70.62 | 11.25 | 8.75 | 12.50 | 21.25 | 0.00 | 1.25 |
| 581 | 53.75 | 65.62 | 12.50 | 11.88 | 6.25 | 16.88 | 0.62 | 0.62 |

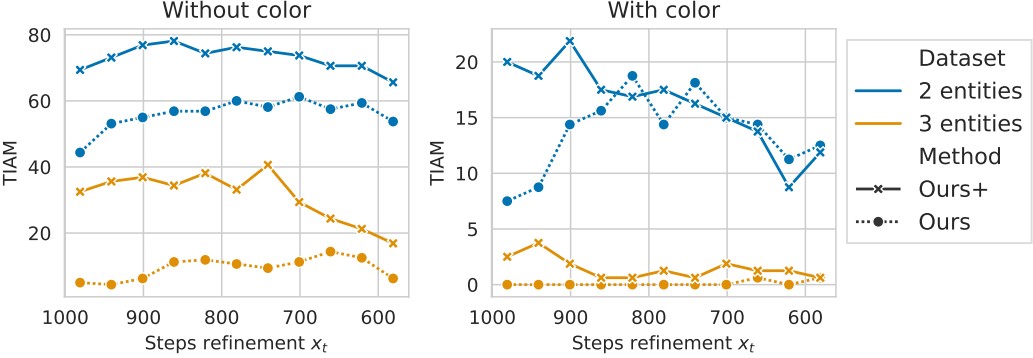

Figure 27: TIAM score of datasets of 10 prompts with 2 and 3 objects as a function of the refinement step used. On the right, entities are bound with colors, we then use the TIAM score with color ground truth.

### A.7.2 Test set

For our methods, we report the TIAM score according to the iterative refinement steps used for the line plot in the main paper, as shown in Table 15. Additionally, we present the CLIP Score (Figure 30) and Aesthetic Score (Figure 31) corresponding to the refinement steps applied in our methods. Notably, we observe that the Aesthetic Score remains constant regardless of the iterative refinement steps used. Furthermore, we observe similar trends to those reported in the main paper regarding the TIAM score. Specifically, applying iterative refinement at slightly later diffusion steps appears to improve the CLIP score. However, delaying the refinement too much results in a decline in performance over time.

We aggregate the TIAM scores per seed across all datasets and methods, with the results shown in Figure 32. The accuracy of InitNO and InitNO+ is somewhat inflated due to their

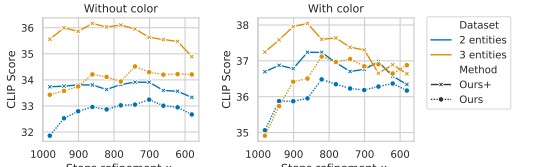
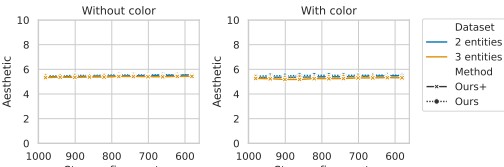

Figure 28: CLIP score according to the iterative refinement step used for the validation datasets.

Figure 29: Aesthetic score according to the iterative refinement step used for the validation. The Aesthetic score is between 1 and 10.

Table 15: TIAM score as a function of different iterative refinement steps.

| step of iterative refinement | 2 entities | | | | 3 entities | | | |
| | w/o colors | | colors | | w/o colors | | colors | |
| | ø | GSNg | ø | GSNg | ø | GSNg | ø | GSNg |
|---|---|---|---|---|---|---|---|---|
| 981 | 58.77 | 78.83 | 7.81 | 19.46 | 13.25 | 43.02 | 0.33 | 2.85 |
| 941 | 62.00 | 81.10 | 8.42 | 20.54 | 17.08 | 45.79 | 0.29 | 2.77 |
| 901 | 65.10 | 81.46 | 9.67 | 20.08 | 20.29 | 47.69 | 0.40 | 2.29 |
| 861 | 65.77 | 81.02 | 9.13 | 19.46 | 22.02 | 49.52 | 0.38 | 2.48 |
| 821 | 65.81 | 80.56 | 8.71 | 18.21 | 23.12 | 48.98 | 0.38 | 2.25 |
| 781 | 66.56 | 79.10 | 8.42 | 16.48 | 25.21 | 48.65 | 0.38 | 1.88 |
| 741 | 66.42 | 78.25 | 8.38 | 15.17 | 25.40 | 47.50 | 0.38 | 1.52 |
| 701 | 65.38 | 77.23 | 8.35 | 12.83 | 24.54 | 44.35 | 0.46 | 1.35 |
| 661 | 64.77 | 74.40 | 7.60 | 11.58 | 23.67 | 41.10 | 0.31 | 0.88 |
| 621 | 63.31 | 71.60 | 7.21 | 10.19 | 23.17 | 38.15 | 0.38 | 0.54 |
| 581 | 61.81 | 69.62 | 6.71 | 8.62 | 22.19 | 33.96 | 0.31 | 0.58 |

multi-round optimization process, where they resample noise if the initial seed does not perform well, leading to artificially improved results. Our best setup, Ours+, consistently achieves higher average scores than our closest competitor, Syngen. We observe greater robustness across seeds, reflected in a lower interquartile range across all datasets, indicating a higher success rate.

Grimal et al. (2024) observed that entities positioned earlier in a prompt tend to appear more frequently than those listed later. In Figure 33, we report the proportion of occurrences of entities based on their position in the prompt. This trend persists across most methods, with the exception of Ours+ and Syngen, particularly for prompts involving two or three colored entities, where this bias is less pronounced.

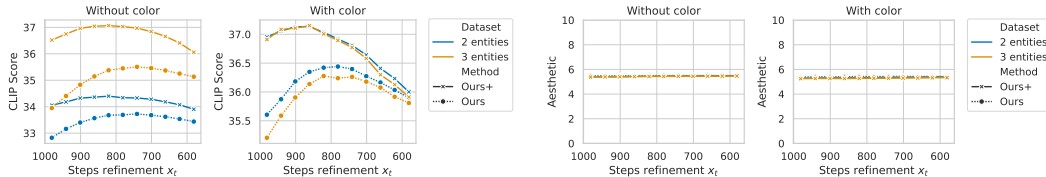

Figure 30: CLIP score according to the iterative refinement step used for the test datasets.

Figure 31: Aesthetic score according to the iterative refinement step used for the validation. The Aesthetic score is between 1 and 10.

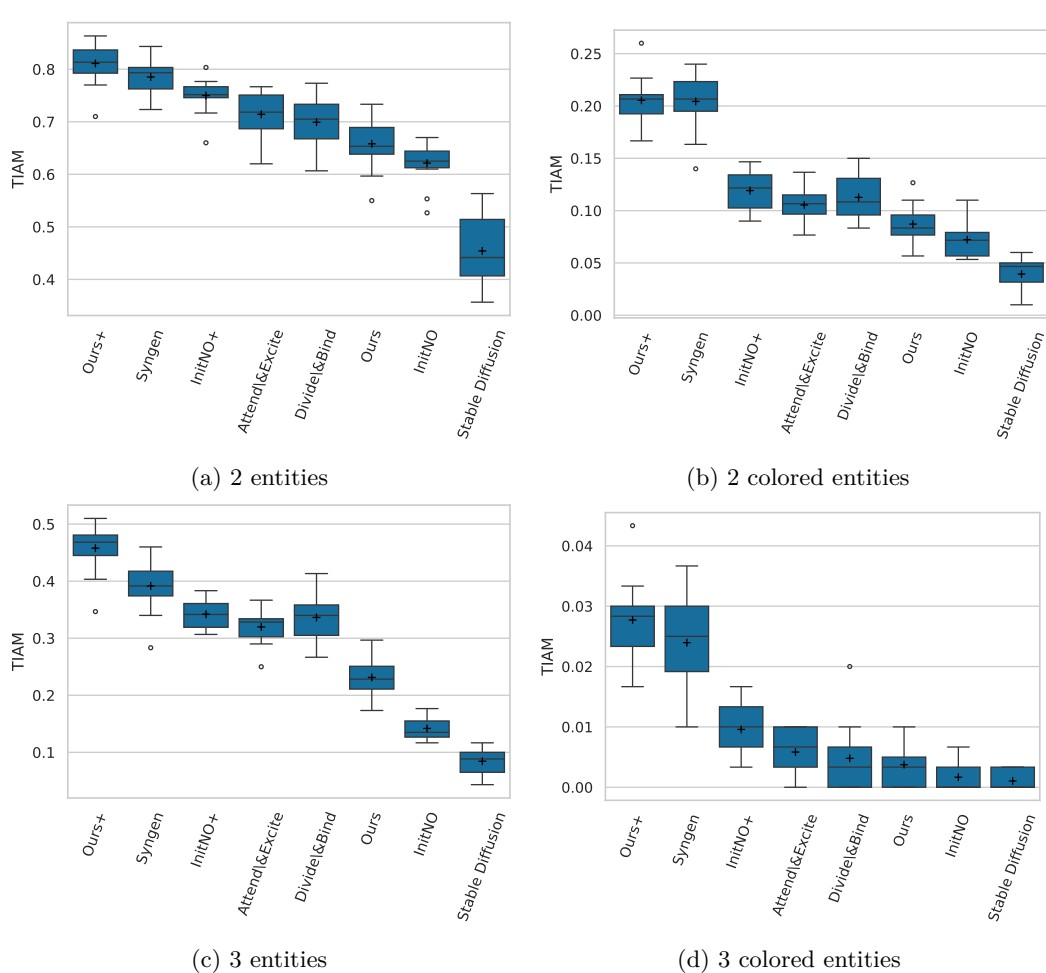

(a) 2 entities

(b) 2 colored entities

(c) 3 entities

(d) 3 colored entities

Figure 32: TIAM aggregate per seed for the 16 seeds per dataset. + shows the mean.

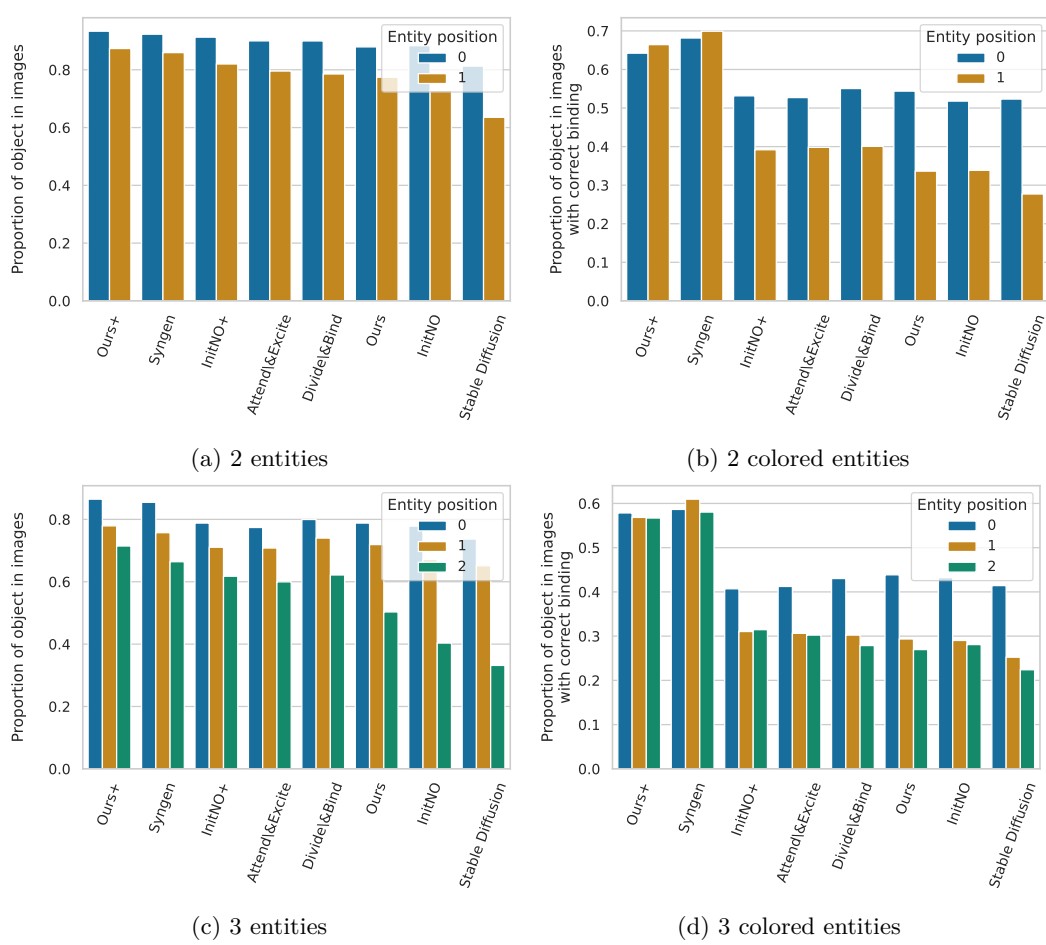

Figure 33: The proportion of occurrences for each entity based on its position within the prompt across all datasets. Here, we focus solely on the detection of entities, regardless of whether their colors are incorrectly attributed.

