# OpenReview forum: "Signal Dynamics in Diffusion Models: Enhancing Text-to-Image Alignment through Step Selection"
_ICLR.cc/2025/Conference — Submitted to ICLR 2025_

### Official Review · Reviewer_LMut · 2024-10-29

**Soundness:** 2
**Presentation:** 3
**Contribution:** 3
**Rating:** 5
**Confidence:** 4

**Summary:**

This paper proposes an approach to improving text-image alignment that optimizes only one specific step during the generation process. Based on the analysis of experimental results, the authors show that later optimization brings better results. Extensive experiments are conducted to demonstrate the effectiveness of the methods.

**Strengths:**

1. The paper is well written.
2. The method is simple, clear, and easy to follow.
3. The experiments are extensive and reproducible.

**Weaknesses:**

1. The qualitative results are insufficient to support the stated improvement:
e.g., in Fig. 1 case "a photo of a car and a blue cat", the attribute blue is still leaked to entity car;
        in Fig. 1 case "a photo of a giraffe and a bear", the bear is still not well generated;
        in Fig. 1 case "a photo of a giraffe and a banana", the number of bananas is wrong;
        in Fig. 6 case "a photo of a giraffe next to a car and a carrot", the carrots and the giraffe are mixed;
        in Fig. 6 case "a photo of a refrigerator next to a horse and a car", the location of each object is not reasonable compared with other methods.
2. The proposed method is incremental and lacks novelty, compared with Attend-and-Excite[1] and A-star[2].
3. The quantitative improvement is marginal in most widely used metrics. The method outperforms others only in terms of TIAM[3], which is not a well-recognized evaluation metric yet.
4. The experiments are conducted on SD 1.4, which falls behind current advanced models like SDXL, SD 3, and Pixart. Perhaps, the phenomenon in SD 1.4 and SDXL are different. The outdated base model makes this research less practical.

[1]Hila Chefer, Yuval Alaluf, Yael Vinker, Lior Wolf, and Daniel Cohen-Or. Attend-and-excite: Attention-based semantic guidance for text-to-image diffusion models. ACM Trans. Graph., 42(4), jul 2023. ISSN 0730-0301. doi: 10.1145/3592116. URL https://doi.org/10.1145/3592116.
[2] Aishwarya Agarwal, Srikrishna Karanam, K J Joseph, Apoorv Saxena, Koustava Goswami, and Balaji Vasan Srinivasan. A-star: Test-time attention segregation and retention for text-to-image synthesis. In Proceedings of the IEEE/CVF International Conference on Computer Vision (ICCV), pp. 2283–2293, October 2023.
[3]Paul Grimal, Hervé Le Borgne, Olivier Ferret, and Julien Tourille. Tiam - a metric for evaluating alignment in text-to-image generation. In Proceedings of the IEEE/CVF Winter Conference on Applications of Computer Vision (WACV), pp. 2890–2899, January 2024.

**Questions:**

1. See Weakness 4. It will be better if some experimental results on more recent models are provided.
2. I would like to see the comparison of each method in terms of computational efficiency.
3. The paper is not well organized enough and the authors might consider modifying the figures and tables to better present their work.

---

> ### Author Response · Authors · 2024-11-27
>
> Thank you for your thorough review. We have addressed you concerns below. We have highlighted the changes in the paper in yellow to help identify them.
>
> **Weakness 1**
> Our approach primarily focuses on mitigating catastrophic neglect and improving attribute binding, ensuring that the generated scene aligns more closely with the requested semantics. However, tasks such as reasoning, counting, and spatial arrangement fall outside the scope of our current work, as they rely heavily on the inherent knowledge and reasoning capabilities of the underlying model. While we aim to enhance the global coherence and semantic accuracy of the scene, addressing challenges like object counts or spatial relations would require additional optimization techniques. For instance, a GSN-based method specifically designed to handle object enumeration could potentially address the issue of incorrect counts. However, this is beyond the objectives of our current study. We acknowledge that these limitations point to opportunities for future work and have clarified this in the limitations.
>
> **Weakness 3** Our method demonstrates improvements across all metrics used in prior works, including the similarity score, aesthetic predictor, TIAM (Semantic Object Accuracy Score), and CLIP score. While the quantitative differences in some widely used metrics, such as the similarity score, may appear marginal, this is consistent with previous approaches in the field, which also achieved publication despite similar trends in results. Additionally, while TIAM may not yet be a widely recognized evaluation metric, it specifically addresses semantic object accuracy, which aligns directly with our focus on resolving catastrophic neglect and attribute binding—areas that are not fully captured by other metrics.
>
> **Weakness 4 and Question 1.** We acknowledge the concern regarding the use of Stable Diffusion 1.4. To address this, we have extended our method to the more advanced Stable Diffusion 3 [1] (SD3), a flow-matching model. Detailed implementation steps and experimental results are included in the revised manuscript and Appendix including the automatic metrics and qualitative results. Please note that the extraction and use of attention maps in Stable Diffusion 3 is not straightforward and, as far as we know, we are the first to use them.
>
> **Weakness 2** Please see the response to weakness 1 of the MBpe reviewer, where we discuss the lack of novelty.
>
> **Question 2** We appreciate your suggestion. In response, we have added a computational efficiency comparison between the methods. All methods are identical in terms of operations, differing only in the number of times the latent image is updated. To reflect this, we report the maximum number of latent image updates (worst-case scenario) for each method in Table 1. This metric provides a clear comparison of computational costs.
>
> **Question 3**  Thank you for your feedback. Do you have any specific recommendations to improve the organization for smoother readability?
>
> [1] Patrick Esser, Sumith Kulal, Andreas Blattmann, Rahim Entezari, Jonas Müller, Harry Saini, Yam Levi, Dominik Lorenz, Axel Sauer, Frederic Boesel, Dustin Podell, Tim Dockhorn, Zion English, Kyle Lacey, Alex Goodwin, Yannik Marek, & Robin Rombach. (2024). Scaling Rectified Flow Transformers for High-Resolution Image Synthesis.

---

### Official Review · Reviewer_MBpe · 2024-11-03

**Soundness:** 2
**Presentation:** 2
**Contribution:** 1
**Rating:** 3
**Confidence:** 4

**Summary:**

This paper introduce Signal Dynamics in Diffusion Models, which explore the key stage of signal modifications that could help to yield superior results.

**Strengths:**

1. The method selectively enhance the signal at a key diffusion step, optimizing image generation.
2. Explore and find the better stage to apply signal modification that leads to better results.

**Weaknesses:**

1. The paper lacks novelty. The main concept of the paper is to discover the best step to perform IterRef. It is not new and there are similar discoveries in FreeDoM[1].

2. The paper lacks the comparison between different methods, there are many other methods that could also achive higher text-alignment like RPG[2], SLD[3], the results in the paper are not competitive enough to support the claim of sota.

3. The current experimental analysis also appears insufficient. More evaluation metrics like FID,IS,T2I-CompBench[4] should be used to provide a more comprehensive results.

[1] Yu, Jiwen, et al. "Freedom: Training-free energy-guided conditional diffusion model." Proceedings of the IEEE/CVF International Conference on Computer Vision. 2023.

[2] Yang, Ling, et al. "Mastering text-to-image diffusion: Recaptioning, planning, and generating with multimodal llms."

[3] Wu, Tsung-Han, et al. "Self-correcting llm-controlled diffusion models."

[4] Huang, Kaiyi, et al. "T2i-compbench: A comprehensive benchmark for open-world compositional text-to-image generation." Advances in Neural Information Processing Systems 36 (2023): 78723-78747.

**Questions:**

1. Can this method applied to different architectures like current sota models SD3 or FLUX ?

2. More evaluation results are needed between similar methods.

3. What is the fundemental difference between your findings and the results showed in Fig.3 of FreeDoM ?

---

> ### Author Response · Authors · 2024-11-27
>
> Thank you for your comments and the time spent reviewing the paper. We have responded to your feedback below. We have highlighted in yellow in the paper the changes for quick identification.
>
> **Weakness 1 and Question 3**
> Thank you for pointing out this paper. The identification of distinct periods during image construction is indeed not new, and we have cited related works [1, 2] in our paper. Upon reviewing FreeDoM, we note that their approach is more closely aligned with applying Classifier Guidance using an external model. For example, in their [textual guidance implementation](https://github.com/vvictoryuki/FreeDoM/blob/1394b1dc5807fb01db6a26d2dc42ca05e3d2eaf5/Face-GD/functions/denoising.py#L39), they apply 10 updates of the latent image at each sampling step between steps 800 and 500, while performing a single update at other steps. This method can be interpreted as performing IterRef between steps 800 and 500 and applying GSN guidance at other steps. Moreover, FreeDoM requires resampling an $x_t$ at each step, rather than directly modifying the latent image. Our approach, in contrast, selects a single step for IterRef and demonstrates that previous methods can achieve better results with improved choices of IterRef steps. Finally Freedom do not relies on the inherent knowledge of the model and need external knowledge.
>
> **Weakness 2**  In our paper, we focus on comparing methods that rely solely on the internal knowledge of the diffusion model to enhance generation, without leveraging external models or additional training. The approaches mentioned rely on external components, such as large language models (LLMs), to guide the generation process. Our claim of performance is specifically made within the context of training-free methods that rely only on the diffusion model's internal mechanisms for adjusting inference. We have clarified this distinction in the paper.
>
> **Weakness 3 and Question 2** We follow prior works such as Attend&Excite [3], InitNO [4], Divide&Bind [5], and A-Star [6] by using the similarity score and a user study. Additionally, we include metrics like the aesthetic predictor, TIAM (Semantic Object Accuracy Score), and CLIP score to provide a more comprehensive evaluation compared to previous approaches. Regarding T2I-CompBench, it focuses on tasks like counting, which are beyond the scope of our work, which addresses catastrophic neglect and attribute binding. Similarly, FID and IS are general quality metrics that remain largely unchanged since we use the same model. The *aesthetic score* reported in Table 2 shows that the image global quality is almost the same in all experiments (it varies from 5.3 to 5.5). As specified in the introduction (title and abstract too) we specifically aim to improve text-image alignment in these challenging aspects.

---

> ### Author Response · Authors · 2024-11-27
>
> **Question 1**
> This approach can indeed be applied to recent SOTA models such as Stable Diffusion 3 (SD3) [7] and FLUX. However, to the best of our knowledge, GSN has never been applied to the Stable Diffusion 3 architecture and it is therefore necessary to make appropriate choices with regard to certain implementation details, as the extraction of attention maps is not straightforward. Following the remark of the reviewer, we extended the proposed method to SD3, a flow-matching model, and provide detailed explanations on its implementation in Appendix. Since the approach is fundamentally based on the iterative reconstruction of an image, it is compatible with models that follow a similar process. Specifically, we leverage the features of SD3 by extracting the attention between text and visual features to guide the generation process. This allows us to identify an optimal step to enhance the signal. In the revised manuscript, we have included scores for this model and detailed the procedures for feature extraction and processing in the Appendix. Additionally, we have provided qualitative comparisons to further illustrate the effectiveness of the approach in Appendix.
>
> [1] Choi, J., Lee, J., Shin, C., Kim, S., Kim, H., & Yoon, S. (2022). Perception Prioritized Training of Diffusion Models. In _Proceedings of the IEEE/CVF Conference on Computer Vision and Pattern Recognition (CVPR)_ (pp. 11472-11481).
>
> [2] Ting Chen. (2023). On the Importance of Noise Scheduling for Diffusion Models.
>
> [3] Chefer, H., Alaluf, Y., Vinker, Y., Wolf, L., & Cohen-Or, D. (2023). Attend-and-Excite: Attention-Based Semantic Guidance for Text-to-Image Diffusion Models_. ACM Trans. Graph., _42_(4).
>
> [4] Xiefan Guo, Jinlin Liu, Miaomiao Cui, Jiankai Li, Hongyu Yang, & Di Huang. (2024). InitNO: Boosting Text-to-Image Diffusion Models via Initial Noise Optimization.
>
> [5] Li, Y., Keuper, M., Zhang, D., & Khoreva, A. (2023). Divide & bind your attention for improved generative semantic nursing. In _34th British Machine Vision Conference 2023, BMVC 2023_.
>
> [6] Agarwal, A., Karanam, S., Joseph, K., Saxena, A., Goswami, K., & Srinivasan, B. (2023). A-STAR: Test-time Attention Segregation and Retention for Text-to-image Synthesis. In _Proceedings of the IEEE/CVF International Conference on Computer Vision (ICCV)_ (pp. 2283-2293).
>
> [7] Patrick Esser, Sumith Kulal, Andreas Blattmann, Rahim Entezari, Jonas Müller, Harry Saini, Yam Levi, Dominik Lorenz, Axel Sauer, Frederic Boesel, Dustin Podell, Tim Dockhorn, Zion English, Kyle Lacey, Alex Goodwin, Yannik Marek, & Robin Rombach. (2024). Scaling Rectified Flow Transformers for High-Resolution Image Synthesis.

---

### Official Review · Reviewer_3QPk · 2024-11-04

**Soundness:** 2
**Presentation:** 2
**Contribution:** 2
**Rating:** 5
**Confidence:** 3

**Summary:**

The paper titled "Signal Dynamics in Diffusion Models: Enhancing Text-to-Image Alignment through Step Selection" discusses a method to improve text-to-image alignment in generative AI models by enhancing signals at critical diffusion steps based on input semantics.  The study also highlights the importance of selecting the right diffusion steps for signal enhancement.

**Strengths:**

1. The assessment is comprehensive, indicating a thorough evaluation.
2. The performance of the proposed method surpasses that of previous work, as demonstrated by the comparative results presented in the paper's table.

**Weaknesses:**

1. The source code is inaccessible via the anonymous link provided.
2. The paper introduces numerous concepts without adequate explanation, which complicates comprehension:
* The term "GSN guidance" is not clearly defined. Is it a concept coined by the authors? If "GSN guidance" refers to the optimization of latents, why isn't the proposal method considered a form of "GSN guidance"? What is the distinction between "Ours" and "Ours+"?
* The phrase "lacks inherent semantic meaning" is used but not further elaborated upon.
* It is unclear what the "Similarity Score" is intended to measure.
3. The paper's novelty is questionable. The optimal iteration appears to be simply a result of the author's experimentation with various step iterations. But it has been widely accepted in research that detailed results are typically generated after the 8th step.

**Questions:**

Refer to the Weaknesses.

---

> ### Author Response · Authors · 2024-11-27
>
> Thank you for your insightful comments and time spent reviewing the paper. We have addressed your concerns below and have highlighted the change in the paper for easy reference.
>
> **Weaknesses**
> 1. We acknowledge that the provided link was a placeholder and not a functional link to the source code. To clarify, we have updated the paper to explicitly state, "The code will be released upon publication of the paper."
> 2. Clarification
> - Attend&Excite [1] introduces iterative refinement (line 9 of their algorithm), where the latent space is updated multiple times at specific steps of the diffusion process. For all other steps, the latent space is updated only once per step. To distinguish between these two processes, we define a single application of this iterative refinement as GSN guidance in our work. As explained in the paper, GSN guidance can be repeated at selected diffusion steps, which we term iteref—referring to multiple applications of GSN guidance (i.e., several updates to the latent space within a single step). "Ours" refers to applying a single step of iterative refinement (without additional GSN guidance), while "Ours+" involves applying a single step of refinement followed by continued denoising using iterative refinement.
> - On the phrase "lacks inherent semantic meaning": We have revised this phrase in the paper for clarity. As explained by Kwon et al. [2], direct manipulation of the image in the latent space is not feasible. Instead, the semantic latent space within the diffusion model must be leveraged to guide changes in the latent space.
> - About the Similarity Score: This metric, introduced by Attend&Excite and subsequently used by GSN-related works (e.g., Divide&Bind [4], InitNO [3], and A-Star [5]), measures the semantic similarity between the generated image and the prompt. It includes three components: *Full Prompt Similarity* the similarity between the prompt and the image embedding.
> *Minimum Object Similarity* the minimum similarity between individual textual entities and the image.
> *Text-Text Similarity* the similarity between the generated image's caption and the prompt. Due to space constraints, the main text does not include full metric details, but these are provided in the appendix. To improve clarity, we now explicitly state in the paper that this metric is referred to as the Similarity Score.
> 3. Our work builds on the findings of [6, 7], as cited in the paper, which highlight that the diffusion process consists of multiple step windows during which the model constructs different aspects of the image. Consequently, refining the entire diffusion process is not necessarily effective. Instead, improvements can be achieved more efficiently by targeting specific steps where signal recovery is most impactful. Previous approaches lacked a systematic explanation for selecting refinement steps or determining the optimal number of refinements, often relying solely on empirical results without deeper theoretical justification. To address this gap, we demonstrate that refinement steps should be carefully selected based on signal recovery dynamics. Furthermore, we propose a lightweight approach to efficiently identify the most critical steps for refinement, ensuring a more principled and effective process.
>
> [1] Chefer, H., Alaluf, Y., Vinker, Y., Wolf, L., & Cohen-Or, D. (2023). Attend-and-Excite: Attention-Based Semantic Guidance for Text-to-Image Diffusion Models_. ACM Trans. Graph., _42_(4).
>
> [2] Mingi Kwon, Jaeseok Jeong, & Youngjung Uh. (2023). Diffusion Models already have a Semantic Latent Space.
>
> [3] Xiefan Guo, Jinlin Liu, Miaomiao Cui, Jiankai Li, Hongyu Yang, & Di Huang. (2024). InitNO: Boosting Text-to-Image Diffusion Models via Initial Noise Optimization.
>
> [4] Li, Y., Keuper, M., Zhang, D., & Khoreva, A. (2023). Divide & bind your attention for improved generative semantic nursing. In _34th British Machine Vision Conference 2023, BMVC 2023_.
>
> [5] Agarwal, A., Karanam, S., Joseph, K., Saxena, A., Goswami, K., & Srinivasan, B. (2023). A-STAR: Test-time Attention Segregation and Retention for Text-to-image Synthesis. In _Proceedings of the IEEE/CVF International Conference on Computer Vision (ICCV)_ (pp. 2283-2293).
>
> [6] Choi, J., Lee, J., Shin, C., Kim, S., Kim, H., & Yoon, S. (2022). Perception Prioritized Training of Diffusion Models. In _Proceedings of the IEEE/CVF Conference on Computer Vision and Pattern Recognition (CVPR)_ (pp. 11472-11481).
>
> [7] Ting Chen. (2023). On the Importance of Noise Scheduling for Diffusion Models.

---

### Meta-Review · Area_Chair_nwBH · 2024-12-17

**Metareview:**

This work focuses on improving text-image alignment by optimizing one specific step in diffusion process. Its strength includes not treating every step equal and training-free. However, reviewers all agree that the novelty is limited and two of them have major concerns on insufficient evaluation. It receives one clear reject and two borderline reject. The limited novelty embodies in the similarity with a few existing works explicitly pointed out by reviewers and insufficient evaluation refers to lack of comparison with counterparts and lack of evaluation using mainstream metrics. The rebuttal unfortunately did not address those concerns. By taking all reviews and discussions into account, a decision of reject is made. Authors are suggested to continue improving it by incorporating reviewers' suggestion and resubmit elsewhere.

**Additional Comments On Reviewer Discussion:**

The two main points raised by reviewers are limited novelty and insufficient evaluation. In rebuttal, authors gave some explanations about the difference with prior work and add a few more experimental results. However, reviewers still keep their original score and after going over the details, AC agrees that those are unconvincing to address raised concerns, especially the difference with prior work (i.e., the contribution) is still below the conference bar, and thus made the reject decision.

---

### Decision · Program_Chairs · 2025-01-22

Reject